# The role of sigma 1 receptor in organization of endoplasmic reticulum signaling microdomains

Vladimir Zhemkov[1], Jonathon A Ditlev[2†‡], Wan-Ru Lee[1], Mikaela Wilson[1], Jen Liou[1], Michael K Rosen[2], Ilya Bezprozvanny[1,3]*

[1]Department of Physiology, UT Southwestern Medical Center at Dallas, Dallas, United States; [2]Department of Biophysics, Howard Hughes Medical Institute, UT Southwestern Medical Center at Dallas, Dallas, United States; [3]Laboratory of Molecular Neurodegeneration, Peter the Great St. Petersburg State Polytechnic University, St. Petersburg, Russian Federation

**Abstract** Sigma 1 receptor (S1R) is a 223-amino-acid-long transmembrane endoplasmic reticulum (ER) protein. S1R modulates activity of multiple effector proteins and is a well-established drug target. However, signaling functions of S1R in cells are poorly understood. Here, we test the hypothesis that biological activity of S1R in cells can be explained by its ability to interact with cholesterol and to form cholesterol-enriched microdomains in the ER membrane. By performing experiments in reduced reconstitution systems, we demonstrate direct effects of cholesterol on S1R clustering. We identify a novel cholesterol-binding motif in the transmembrane region of human S1R. Mutations of this motif impair association of recombinant S1R with cholesterol beads, affect S1R clustering in vitro and disrupt S1R subcellular localization. We demonstrate that S1R-induced membrane microdomains have increased local membrane thickness and that increased local cholesterol concentration and/or membrane thickness in these microdomains can modulate signaling of inositol-requiring enzyme 1α in the ER. Further, S1R agonists cause disruption of S1R clusters, suggesting that biological activity of S1R agonists is linked to remodeling of ER membrane microdomains. Our results provide novel insights into S1R-mediated signaling mechanisms in cells.

*For correspondence:
ilya.bezprozvanny@
utsouthwestern.edu

Present address: †Department of Biochemistry, University of Toronto, Toronto, Canada; ‡Program in Molecular Medicine, Hospital for Sick Children, Toronto, Canada

**Competing interests:** The authors declare that no competing interests exist.

## Introduction

Cholesterol is an essential component of cellular membranes, and levels of cholesterol in cells are tightly controlled (*Brown and Goldstein, 1999*; *Goldstein and Brown, 2015*; *Radhakrishnan et al., 2008*). The plasma membrane (PM) is highly enriched in cholesterol and other sterols, containing 30–40% cholesterol and 10–30% sphingolipids based on molar amounts (*van Meer et al., 2008*). It is widely accepted that cholesterol is not distributed homogeneously in the PM and instead forms lipid rafts, or cholesterol-enriched microdomains, that play important signaling roles in cells (*Levental et al., 2020*). Although cholesterol is produced in the endoplasmic reticulum (ER), the concentration of cholesterol in ER membranes is much lower than in the PM (*van Meer et al., 2008*). Level of ER cholesterol is maintained by an "on-and-off" switch of the Scap-SREBP pathway with a half-maximal response of SREBP-2 at about 5 molar % cholesterol in the ER (*Radhakrishnan et al., 2008*). There is limited information about spatial distribution of cholesterol in the ER membrane, but recent studies suggested existence of cholesterol-enriched microdomains in the ER membrane (*Area-Gomez et al., 2012*; *Hayashi and Fujimoto, 2010*; *King et al., 2020*; *Montesinos et al., 2020*). One of the most studied examples of such microdomains are mitochondria-associated membranes (MAMs), which are sites on the ER membrane in immediate proximity to the mitochondrial outer membrane (*Csordás et al., 2018*; *Prinz et al., 2020*). MAMs have been

shown to play important roles in a variety of cellular functions, such as ER to mitochondria $Ca^{2+}$ transfer (*Csordás et al., 2018*; *Hajnóczky et al., 2002*), ATP production (*Hajnóczky et al., 2002*), lipid metabolism (*Csordás et al., 2018*; *Vance, 2014*), and autophagy (*Garofalo et al., 2016*). Unsurprisingly, MAM dysregulation was observed in numerous pathophysiological conditions, including neurodegenerative diseases (*Schon and Area-Gomez, 2013*), cancers (*Morciano et al., 2018*), and lysosomal disorders (*Annunziata et al., 2018*; *Sano et al., 2009*).

Sigma 1 receptor (S1R) is a 223-amino-acid-long, single-pass transmembrane ER protein (*Hayashi, 2019*; *Ryskamp et al., 2019*; *Schmidt and Kruse, 2019*) that has a short cytoplasmic tail and a large luminal ligand-binding domain (*Mavylutov et al., 2018*; *Schmidt et al., 2016*). It has been suggested that S1R acts as a molecular chaperone that can stabilize a native conformation of multiple client proteins in stress conditions (*Hayashi, 2019*; *Nguyen et al., 2017*; *Su et al., 2010*). Recent studies also suggested that S1R can act as an RNA-binding protein (*Lee et al., 2020*). S1R is highly enriched in the liver and expressed in the nervous system. Analysis of S1R knockout (KO) mice revealed a number of nervous system abnormalities (*Couly et al., 2020*), suggesting that S1R plays an important role in neurons. In humans, mutations in S1R lead to a juvenile form of amyotrophic lateral sclerosis (ALS) (*Al-Saif et al., 2011*) and distal hereditary motor neuropathies (dHMNs) (*Almendra et al., 2018*; *Gregianin et al., 2016*; *Horga et al., 2016*; *Li et al., 2015*; *Ververis et al., 2020*). In animal models, genetic ablation of S1R exacerbates pathology of several neurological disorders (*Hong et al., 2017a*; *Mavylutov et al., 2013*; *Wang et al., 2017*). S1R is considered to be a potential drug target for treatment of neurodegenerative disorders and cancer (*Herrando-Grabulosa et al., 2021*; *Kim and Maher, 2017*; *Ryskamp et al., 2019*). S1R binds multiple classes of drugs with nano- and sub-micromolar affinities (*Maurice and Su, 2009*). Based on their biological activity, S1R ligands are classified into agonists and antagonists (*Maurice and Su, 2009*). Signaling functions of S1R in cells are under intense investigation. The most prominent hypothesis is that under resting conditions S1R forms an inert complex with GRP78/BiP protein in the ER (*Hayashi and Su, 2007*). When activated by agonists or in conditions of cellular stress, S1R dissociates from BiP and is able to interact with a variety of client proteins (*Hayashi and Su, 2007*).

S1R preferentially localizes to MAMs (*Hayashi and Su, 2007*), and genetic KO of S1R in mice results in impaired MAM stability as evidenced by reduced number of contacts on electron micrographs and biochemical fractionation of MAM components (*Watanabe et al., 2016*). MAMs are cholesterol-enriched microdomains within the ER membrane (*Area-Gomez et al., 2012*; *Hayashi and Fujimoto, 2010*; *Montesinos et al., 2020*), and previous studies demonstrated that S1R can directly interact with cholesterol and ceramides in MAMs (*Hayashi and Fujimoto, 2010*; *Hayashi and Su, 2004*; *Palmer et al., 2007*). In the present study, we tested the hypothesis that S1R association with cholesterol plays a critical role in organization of specialized lipid microdomains in the ER, including MAMs. Using two in vitro reconstitution systems, we demonstrated cholesterol-dependent clustering of S1R in lipid bilayers. We identified a novel cholesterol-binding site in the S1R sequence and demonstrated the importance of this site for S1R clustering. We further demonstrated that S1R agonists reduced the number and size of S1R clusters in the presence of cholesterol. Based on these results, we conclude that the main biological function of S1R in cells is related to its ability to organize and remodel cholesterol-enriched microdomains in the ER. Our conclusion is consistent with MAM defects observed in the S1R KO mice (*Watanabe et al., 2016*).

## Results

### S1R localizes to MAMs

To investigate the subcellular localization of S1R, we transfected HEK293 cells with a construct encoding a S1R-GFP fusion protein and performed confocal imaging experiments. Previous reports indicated that intracellular localization of the C-terminally tagged S1R is similar to that of the endogenous receptor (*Hayashi and Su, 2003b*). To visualize the ER, we co-transfected cells with mCherry-Sec61β (*Zurek et al., 2011*). In agreement with previous reports (*Hayashi and Su, 2003a*; *Hayashi and Su, 2003b*), S1R-GFP formed puncta in the ER (*Figure 1A*). To determine if these puncta corresponded to MAMs, we co-stained cells with a mitochondrial marker TOM20 and established that S1R-GFP puncta were frequently found in close opposition to mitochondria (*Figure 1A*). In contrast to wild-type (WT) S1R, an ALS-causing mutant S1R-E102Q (*Al-Saif et al., 2011*) was

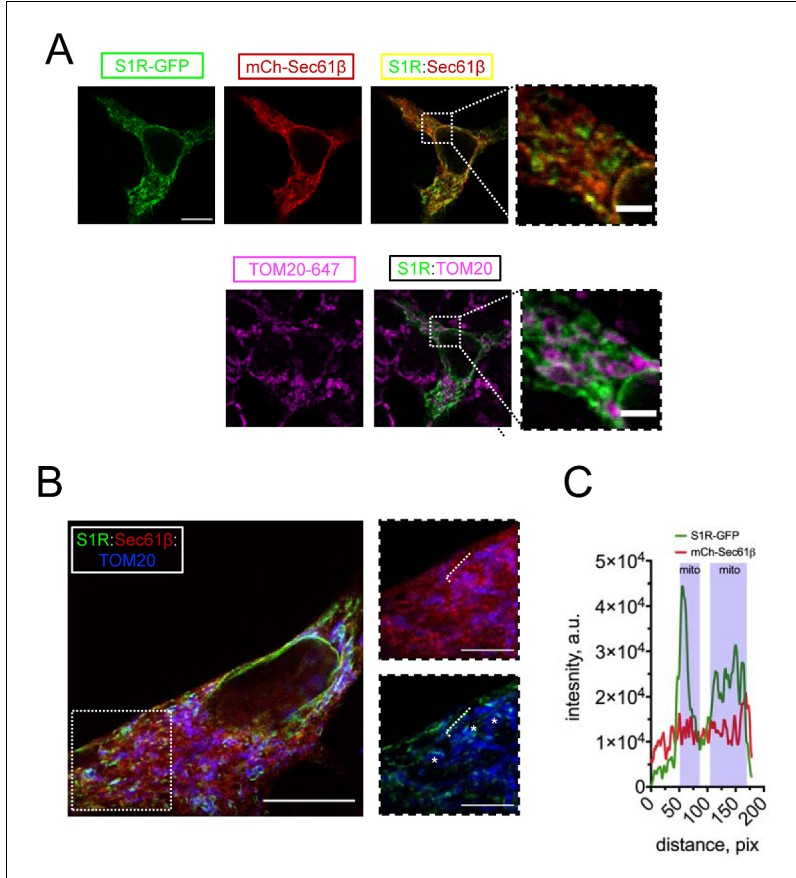

**Figure 1.** S1R targeting to mitochondria-associated membranes (MAMs) in HEK293 cells. (**A**) Intracellular distribution of wild-type S1R-GFP (green) was visualized in HEK293 cells by co-expressing an endoplasmic reticulum (ER) marker mCherry-Sec61β (red) and by immunostaining with a mitochondrial marker TOM20 (magenta). Scale bar = 10 µm; insets = 2.5 µm. (**B**) Specimen was processed according to the protein retention expansion microscopy procedure, with S1R-GFP in green, mCherry-Sec61β in red, and TOM20 staining in blue. Insets show double staining of mCherry-Sec61β and TOM20 (top), S1R-GFP and TOM20 (bottom). Putative MAM compartments are labeled with asterisks. Scale bars = 5 µm; insets = 1.5 µm (real space). (**C**) Intensity profiles for S1R-GFP (green) and mCherry-Sec61β (red) channels along the ER tubule (labelled with a white dashed line on **B**). Mitochondrial proximity regions (shaded blue) were identified based on the intensity of TOM20.

The online version of this article includes the following figure supplement(s) for figure 1:

**Figure supplement 1.** Intracellular distribution of the S1R-E102Q mutant.

**Figure supplement 2.** Purification of MAMs from mice liver.

distributed uniformly in the ER and was not enriched in proximity to mitochondria (*Figure 1—figure supplement 1*). In order to increase the spatial resolution of our experiments, we utilized the protein retention expansion microscopy (pro-ExM) procedure that resulted in a 4.0–4.5-fold physical expansion of the specimen (*Tillberg et al., 2016*). HEK293 cells were co-transfected with S1R-GFP, mCherry-Sec61β, and stained with anti-GFP, anti-mCherry, and anti-TOM20 antibodies and processed according to the pro-ExM procedure for imaging (*Figure 1B*). Measurements of signal intensities of mCherry-Sec61b and S1R-GFP markers along the selected ER tubule indicated enrichment of S1R in the areas of mitochondrial proximity (*Figure 1C*). We then calculated the overlap between S1R-GFP, mCherry-Sec61β, and mitochondria. On average, 12 ± 5% of mCherry-Sec61β signal area overlapped with mitochondria, while a higher degree of overlap was observed for S1R-GFP and mitochondria, with fractional area of 31 ± 7% (n [number of cells] = 5, p-value=0.0034). As an alternative approach, we performed a biochemical fractionation of MAMs from mice liver (*Figure 1—figure supplement 2*), which confirmed enrichment of S1R in MAM fraction. Taken together, this data

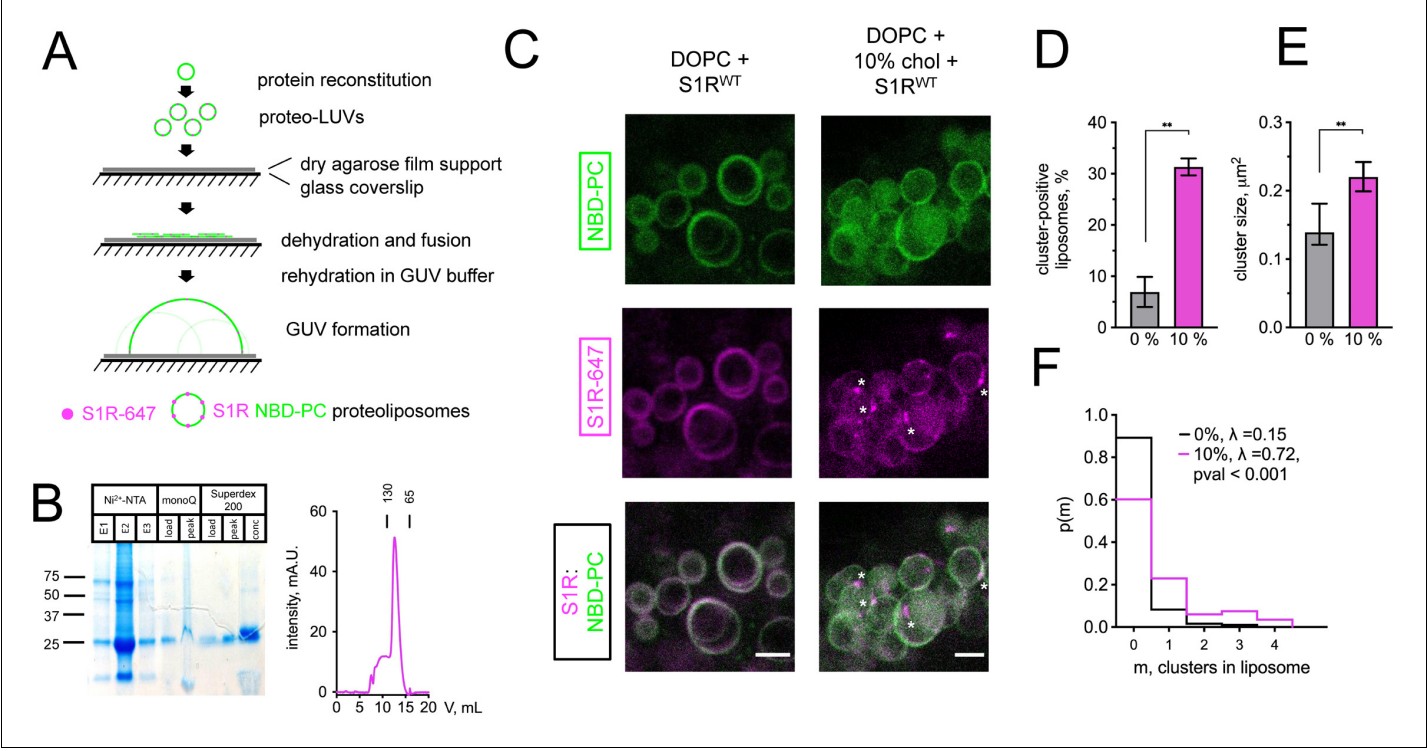

**Figure 2.** Cholesterol-dependent clustering of S1R in giant unilamellar vesicles (GUVs). (**A**) The GUV formation procedure. Purified S1R was labeled with Alexa647 and reconstituted into large unilamellar vesicles (LUVs) to form proteoliposomes. After controlled dehydration of proteoliposomes on an agarose-covered coverslip, films were rehydrated in a salt buffer, which resulted in the formation of micrometer-size GUVs. (**B**) Biochemical characterization of purified S1R by SDS-PAGE analysis (left) and size-exclusion chromatography (right). (**C**) Distribution of S1R-647 in cholesterol-free DOPC liposomes (left) and in the presence of 10 mol % cholesterol (right), with membrane dye NBD-labeled phosphatidylcholine (NBD-PC) shown in green and S1R-647 in magenta. S1R clusters are marked with asterisks. Scale bars = 5 µm. (**D**) Average fraction of cluster-positive liposomes. In each experiment, 7–20 fields of view were quantified blindly. Data is mean ± SEM from n = 3 independent experiments (0% condition: 506 liposomes, 10% condition: 685 liposomes). **p=0.003 based on two-tailed t-test. (**E**) Distribution of the S1R cluster size in GUV in the absence and presence of cholesterol. Data is mean ± 95% CI. **p=0.002 based on two-tailed t-test. (**F**) Distribution of the number of clusters per liposome in the absence (black) and presence of 10 mol % cholesterol (magenta). Data was fitted with Poisson distribution to estimate the average number of clusters per liposome, λ ($N_1$ = 194 liposomes for 0%, $N_2$ = 201 for 10 mol % cholesterol experimental condition). ****p-value<0.001 calculated based on Whitehead's and C-test statistical tests.

The online version of this article includes the following figure supplement(s) for figure 2:

**Figure supplement 1.** Additional examples of S1R distribution in DOPC giant unilamellar vesicles at 0, 10, and 20 mol % cholesterol.

confirmed that S1R-GFP is enriched in ER subdomains in close opposition to mitochondria, consistent with previous reports (*Hayashi and Su, 2007*).

## S1R forms clusters in giant unilamellar vesicles in vitro

To investigate mechanisms responsible for S1R enrichment at ER subdomains, we performed series of in vitro reconstitution experiments with giant unilamellar vesicles (GUVs) (*Figure 2A*). NBD-labeled phosphatidylcholine (NBD-PC) was included in lipid mixtures used to generate GUVs for visualization. Full-length His-tagged S1R was expressed in Sf9 cells and purified to homogeneity by affinity, anion exchange, and gel filtration chromagraphies (*Figure 2B*). After purification, His-S1R protein was covalently labeled with Alexa647 dye via NHS chemistry. To reconstitute S1R in the lipid membrane, large unilamellar vesicles (LUVs) were prepared by extrusion, destabilized with detergent, and mixed with purified S1R. Detergent was subsequently removed using BioBeads. S1R-containing LUVs were dehydrated on an agarose support, and GUVs were formed by rehydration as previously described (*Horger et al., 2009*).

Confocal imaging revealed that when S1R was reconstituted into liposomes composed of 1,2-dioleoyl-sn-glycero-3-phosphocholine (DOPC), it was distributed uniformly (*Figure 2C*, left panel) with

occasional clustering observed at the sites of contact between different GUVs (*Figure 2—figure supplement 1*). In contrast, when vesicles also contained 10% cholesterol (molar ratio to DOPC), S1R was often clustered (*Figure 2C*, right panel, marked with asterisks). As cluster formation is a dynamic process, we made sure to prepare samples simultaneously and image them side by side. On average, S1R clusters were observed in $7 \pm 5\%$ of DOPC liposomes (n = 3, 506 liposomes) and in $31 \pm 3\%$ of DOPC:10 mol % cholesterol liposomes (n = 3, 685 liposomes) (*Figure 2D*). Frequently, the S1R coalesced into one large domain (more examples of S1R behavior in cholesterol-free and cholesterol-containing liposomes are presented in *Figure 2—figure supplement 1*). Quantification of cluster size showed that S1R formed sub-micrometer-sized clusters in GUVs (*Figure 2E*). Cluster size was higher in the presence of cholesterol ($0.16 \pm 0.07$ $\mu m^2$ for DOPC GUVs and $0.27 \pm 0.19$ $\mu m^2$ for cholesterol-containing liposomes) (*Figure 2E*). We used Poisson distribution to compare an average number of S1R clusters per liposome in each condition. We determined that the average number of clusters per liposome was significantly higher for GUVs containing 10 mol % cholesterol ($\lambda = 0.15$ for DOPC GUVs and $\lambda = 0.72$ for cholesterol-containing liposomes, p<0.001, $N_1 = 194$, $N_2 = 201$) (*Figure 2F*). Similar results were obtained when 20 mol % of cholesterol was included in the lipid mixtures used for GUVs formation (data not shown). From these results, we concluded that presence of cholesterol can promote clustering of S1R in the lipid membranes and proposed that association with cholesterol plays a direct role in S1R-induced formation of ER membrane microdomains.

## Cholesterol-binding motifs in the S1R sequence

S1R was shown to interact with cholesterol, other sterols, and sphingolipids in binding studies (*Hayashi and Fujimoto, 2010*; *Hulce et al., 2013*; *Palmer et al., 2007*). Previous studies identified two potential sites of S1R association with cholesterol – Y173 and Y201/Y206 (*Palmer et al., 2007*). However, in the S1R crystal structure (*Schmidt et al., 2016*) Y173 localizes adjacent to the S1R ligand-binding domain and has no membrane contact, and Y201 and Y206 are located within the C-terminal membrane-adjacent amphipathic helices. Additional sequence analysis revealed tandem CARC-like (reverse sequence for the cholesterol-recognition amino acid consensus) motifs R/K-X$_{1-5}$-Y/F/W-X$_{1-5}$-L/V (PLEASE KEEP AMINO ACID SEQUENCE ON 1 LINE) (*Fantini and Barrantes, 2013*) in the transmembrane helix of S1R that span amino acids R7-L14 (*Figure 3A*). To test the importance of this motif, we generated S1R mutants by introducing a GGGG insertion within the CARC motif (S1R-4G) or by mutating W9 and W11 to leucine residues (S1R-W9L/W11L) (*Figure 3A*). We also generated S1R-Y173S and S1R-Y201S/Y206S mutants to test the potential importance of the previously reported cholesterol-binding motifs. In addition to the mutants describe above, we also generated a R7E/R8E mutant or deleted the double arginine motif altogether (ΔRR).

Since mutations introduced in CARC motif are located in close proximity to membrane boundary and can potentially alter protein insertion into the bilayer, we first evaluated their effects on S1R topology. Antibody accessibility studies suggested luminal (*Hayashi and Su, 2007*) or cytoplasmic localization of the C-terminus (*Aydar et al., 2002*), while recent APEX2-enchanced electron microscopy indicated luminal localization of receptor's C-terminus (*Mavlyutov et al., 2017*). We expressed wild-type and mutant S1R-GFP fusions in HEK293 cells and performed selective permeabilization of the PM with digitonin. At low concentrations, digitonin permeabilizes plasma, but not the ER membrane, thus leaving ER luminal epitopes inaccessible to antibodies. When cells expressing wild-type S1R-GFP were stained with anti-GFP antibodies, we did not observe any anti-GFP antibody staining (*Figure 3—figure supplement 1*), indicating that the C-terminus of the S1R-GFP fusion was inaccessible and located in the ER lumen, consistent with previous findings (*Mavlyutov et al., 2017*). Similar results were obtained with S1R-Y173S, S1R-Y201S/Y206S, S1R-4G, and S1R-W9L/W11L mutants (*Figure 3—figure supplement 1*). In contrast, intense anti-GFP antibody staining was observed in experiments with the S1R-R7E/R8E and S1R-ΔRR mutants (*Figure 3—figure supplement 1*), indicating that these proteins had a reversed topology in the ER membrane. Therefore, we excluded these mutants from further analysis.

Wild-type and mutant S1R were expressed in HEK293 cells and their distribution was analyzed by confocal microscopy. Wild-type S1R and the S1R-Y201S/Y206S mutant each formed puncta in the ER (*Figure 3B*). In contrast, the S1R distribution in the ER was diffuse for the S1R-Y173S, S1R-4G, and S1R-W9L/W11L mutants (*Figure 3B*). By performing TOM20-staining experiments, we confirmed that the puncta observed with the wild-type S1R and Y201S/Y206S mutant corresponded to MAMs

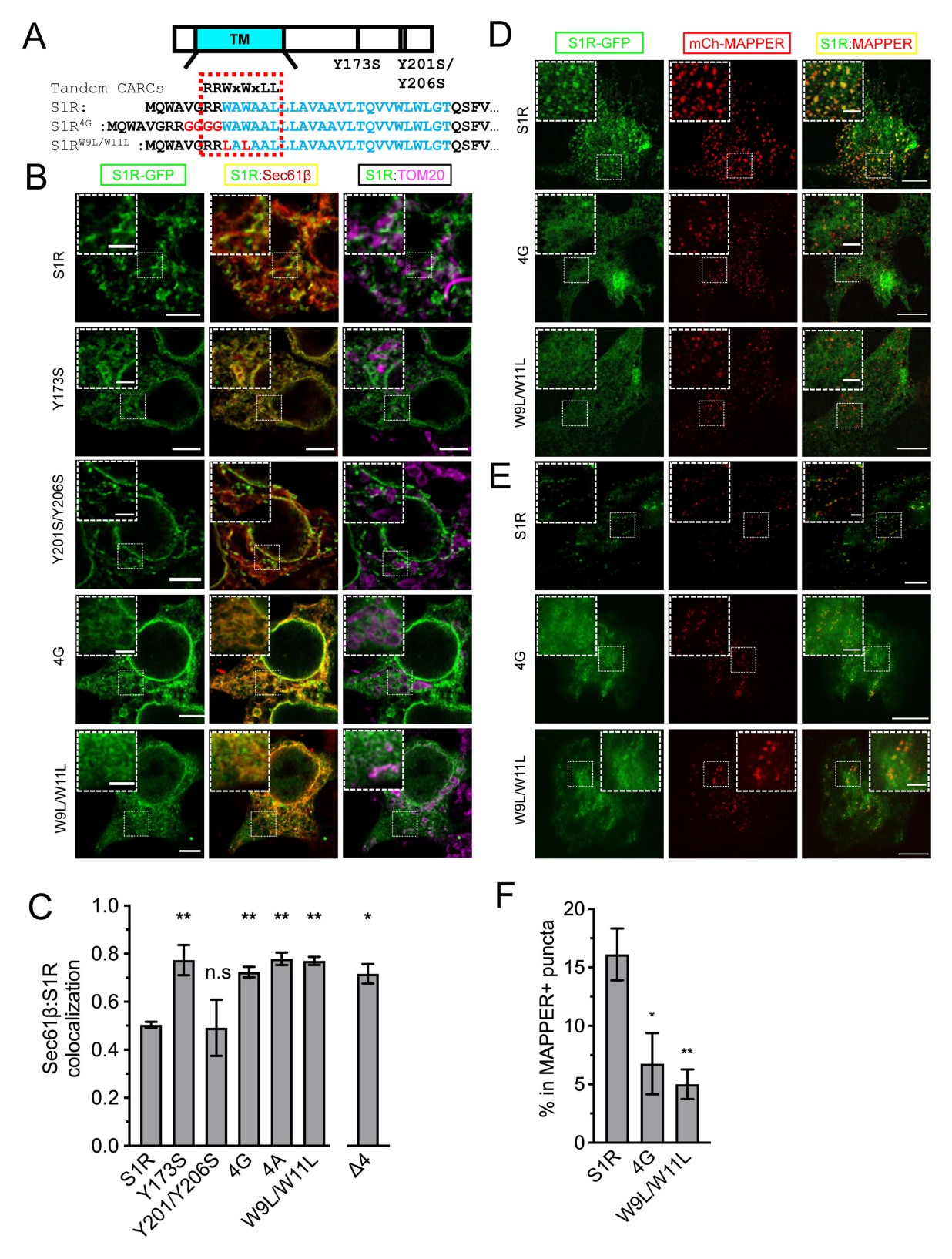

**Figure 3.** Cholesterol-binding motifs in the S1R sequence. (**A**) A schematic representation of the S1R primary sequence with its transmembrane helix (TM) in cyan. Previously proposed cholesterol-binding residues (Y173, Y201/Y206) (**Palmer et al., 2007**) are marked. Sequence analysis identified tandem CARC binding motifs (indicated by a dashed red line) in the TM region (marked in cyan font). Mutations were introduced either by insertion

*Figure 3 continued on next page*

Figure 3 continued

of the four-glycine repeat (4G) or by mutation of the two critical tryptophan residues to leucines (W9L/W11L). (B) Intracellular distribution of the WT receptor and cholesterol-binding mutants in HEK293 cells, S1R-GFP in green, mCherry-Sec61β in red, and anti-TOM20 in magenta. Scale bars = 5 μm, insets = 2.5 μm. (C) Quantification of the Mander's colocalization coefficient between the mCherry-Sec61β and S1R-GFP for WT receptor and mutant forms. Data is mean ± SEM from n = 3 independent experiments (n = 2 for Y173S, Y201S/Y206S, 4A). p-values (n.s. p>0.05, *p<0.05, **p<0.01): Y173S vs. WT: p=0.007, Y201S/Y206S vs. WT: p>0.999, 4G vs. WT: p=0.014, 4A vs. WT: p=0.006, W9L/W11L vs. WT: p=0.003, Δ4 vs. WT: p=0.017, based on ANOVA test with Dunnett's post hoc test. (D) Distribution of the WT S1R-GFP, S1R-4G, and S1R-W9L/W11L mutants (in green) in HEK293 cells in the confocal plane near the plasma membrane (PM). Endoplasmic reticulum (ER)-PM junctions were visualized with a genetically encoded marker MAPPER (in red). Scale bars = 5 μm, insets = 1.5 μm. (E) Distribution of the WT S1R-GFP, S1R-4G, and S1R-W9L/W11L mutants (in green) in HEK293 cells visualized using total internal reflection fluorescence microscopy, with ER-PM junctions labeled with MAPPER (in red). Scale bars = 5 μm, insets = 1.5 μm. (F) Fraction of S1R residing in the MAPPER-positive puncta, calculated from data shown in (E). p-values (*p<0.05, **p<0.01): 4G vs. WT: p=0.025, W9L/W11L vs. WT: p=0.008, based on ANOVA test with Dunnett's post hoc test.

The online version of this article includes the following figure supplement(s) for figure 3:

**Figure supplement 1.** Membrane topologies of the S1R-GFP mutants.
**Figure supplement 2.** Additional characterization of the distribution of S1R-GFP mutants in cells.

(*Figure 3B*). In order to obtain a quantitative measure of the S1R distribution in the ER network, we calculated Mander's overlap coefficient between mCherry-Sec61β and S1R-GFP. Mander's coefficient measures fractional overlap between two channels and varies from 0 (no correlation) to 1 (complete correlation). For wild-type S1R, Mander's coefficient was M = 0.50 ± 0.02 (n = 3, 25 cells), indicating that not all mCherry-Sec61β signal colocalized with the GFP signal in S1R clusters (*Figure 3C*). In contrast, Mander's coefficient was equal to 0.77 ± 0.09 (n = 2, 11 cells) for S1R-Y173S, 0.72 ± 0.04 (n = 3, 30 cells) for S1R-4G, and 0.77 ± 0.03 (n = 3, 17 cells) for S1R-W9L/W11L (*Figure 3C*), reflecting more diffuse distribution patterns. We have also explored the insertion of four alanine stretch (S1R-4A) instead of glycines (*Figure 3C*, *Figure 3—figure supplement 2* for quantification). S1R-4A behaved similarly to S1R-4G mutant. Mander's coefficient for the S1R-Y201S/Y206S mutant was equal to 0.49 ± 0.16 (n = 2, 12 cells) (*Figure 3C*), similar to the wild type. Taken together, we concluded that the newly identified CARC motif is important for S1R targeting to MAMs and that the Y201/Y206 motif is dispensable. Effects of Y173 mutation may be related to misfolding of S1R, as this residue has no membrane contact in the S1R crystal structure (*Schmidt et al., 2016*). To better showcase the distribution of S1R-4G and S1R-W9L/W11L in cells, we performed expansion microscopy of the samples, similar to the experiments that we performed with wild-type S1R (*Figure 1B*). We found that in contrast to the clustered distribution of wild-type S1R, S1R-4G and S1R-W9L/W11L mutants had diffuse localization inside and outside of the ER (*Figure 3—figure supplement 2*).

We then asked if targeting to membrane contact sites is a general property of S1R or it is specific for MAMs. Previously S1R was shown to regulate store-operated Ca$^{2+}$ entry (*Srivats et al., 2016*), and we reasoned that S1R may also be present at ER-PM junctions. To test this hypothesis, we co-expressed S1R-GFP with a genetically encoded ER-PM marker mCherry-MAPPER (*Chang et al., 2013*) in HeLa cells. We first examined subcellular distribution of WT protein and mutants using conventional confocal microscopy near the PM (*Figure 3D*). As reported previously, MAPPER (shown in red) formed small punctate patterns corresponding to the ER membrane in close proximity to the PM (*Figure 3D*). S1R-GFP appeared enriched in these same puncta (*Figure 3D*), in agreement with the previous report (*Srivats et al., 2016*). S1R-4G and S1R-W9L/W11L were distributed diffusely in the ER (*Figure 3D*). When we analyzed cells using total internal reflection fluorescence (TIRF) microscopy, we confirmed enrichment of S1R-GFP in MAPPER-positive puncta compared to the mutants (*Figure 3E*). We quantified S1R-GFP fluorescence in MAPPER-positive puncta and normalized this value to the total GFP fluorescence in the TIRF plane in the same cell. For wild-type S1R-GFP, 16 ± 7% of the fluorescence signal was localized to the ER-PM junctions compared to 7 ± 5% for S1R-4G mutant and 5 ± 3% for S1R-W9L/W11L mutant (*Figure 3F*). Taken together, our results suggest that wild-type S1R localizes to and is enriched in ER contact sites such as MAMs and ER-PM junctions and that CARC motif is important for S1R targeting to these sites.

To further validate the importance of the CARC motif in S1R cholesterol-dependent clustering, we expressed and purified S1R-4G and S1R-W9L/W11L mutant proteins; and performed GUV reconstitution experiments in the presence of 10 mol % cholesterol. The S1R-W9L/W11L mutant formed

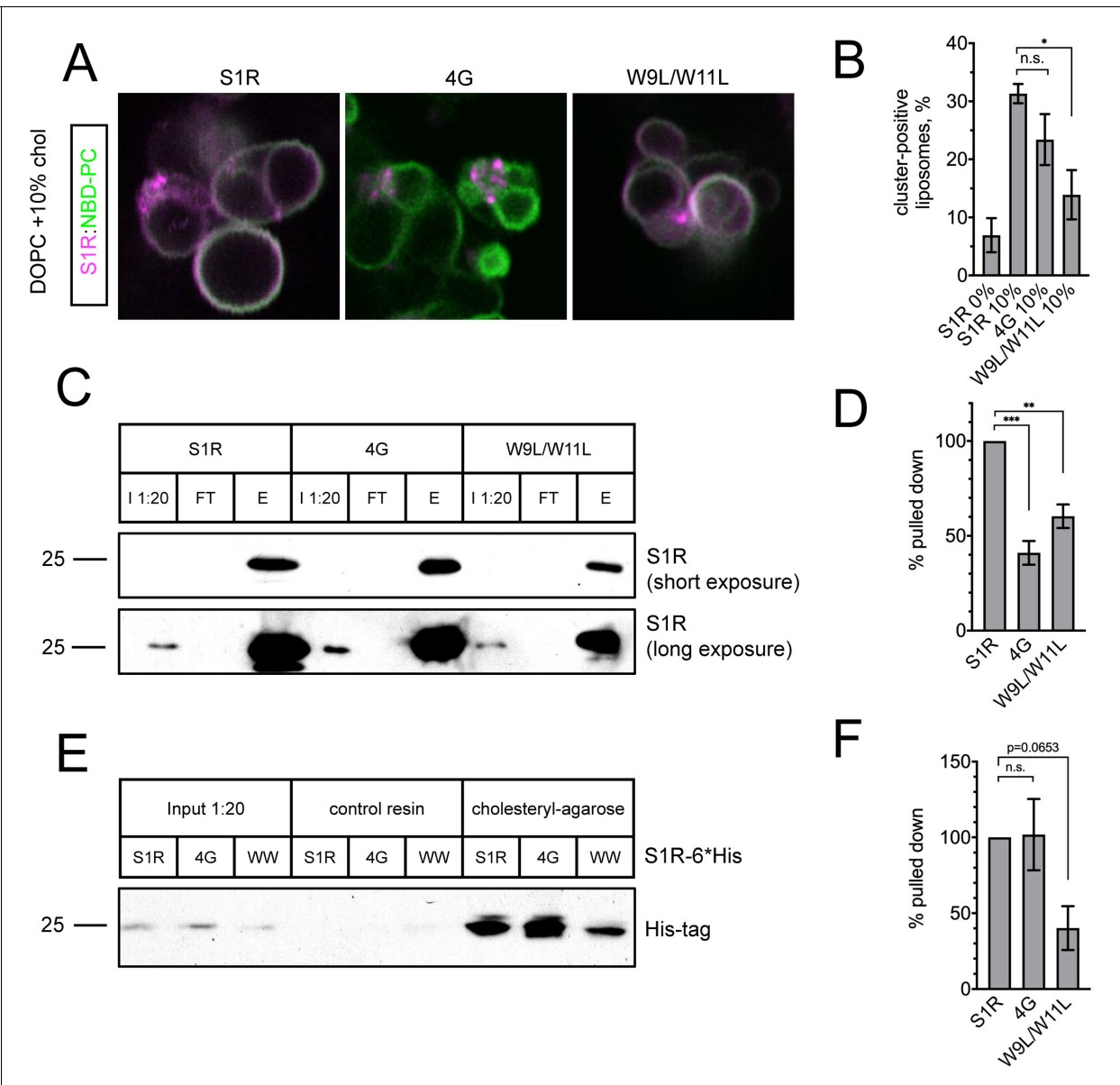

**Figure 4.** Cholesterol binding of the S1R CARC mutants. (**A**) Distribution of purified S1R-4G and S1R-W9L/W11L in cholesterol-containing giant unilamellar vesicles (GUVs) with S1R-647 in magenta and NBD-labeled phosphatidylcholine (NBD-PC) in green. (**B**) Average fraction of cluster-positive liposomes for S1R, S1R-4G, and S1R-W9L/W11L. Cluster-positive liposomes were quantified as in *Figure 2D*. Data (mean ± SEM) is from n = 3 independent experiments (WT 10% condition: 685 liposomes; 4G: 539 liposomes; W9L/W11L: 638 liposomes). p-values (n.s.: non-significant, *p-value<0.05): WT 10% vs. 4G 10%: p=0.31, W9L/W11L 10% vs. WT 10%: p=0.02 based on ANOVA test with Dunnett's post hoc test. (**C**) Cholesterol-coupled agarose pulldown with recombinant S1R protein. I 1:20: input; FT: flow-through; E: eluted protein. Proteins were analyzed with western blot analysis using anti-S1R antibodies. (**D**) Quantification of western blot results shown in (**C**). Data (mean ± SEM) is from n = 3 independent experiments. Measured band intensities of eluted proteins were divided by the measured band intensities of the inputs and normalized to S1R-WT. p-values (**p-value<0.01, ***p-value<0.001): 4G vs. WT: p=0.0003, W9L/W11L vs. WT: p=0.003 based on ANOVA test with Dunnett's post hoc test. (**E**) Cholesterol agarose pulldown with S1R-6His proteins exogenously expressed in HEK293 cells. Proteins were analyzed using western blot analysis with anti-His-tag antibodies. (**F**) Quantification of western blot results shown in (**E**). Data (mean ± SEM) is from n = 3 independent experiments.
Measured band intensities of eluted proteins were divided by the measured band intensities of the inputs and normalized to S1R-WT. p-values: 4G vs. WT: p=0.995, W9L/W11L vs. WT: p=0.065 based on ANOVA test with Dunnett's post hoc test.
The online version of this article includes the following figure supplement(s) for figure 4:

**Figure supplement 1.** Binding of recombinant S1R to control resin (CarboxyLink agarose) and cholesterol-coupled agarose.

less clusters (*Figure 4A, B*), consistent with the diffuse distribution of this mutant in cells (*Figure 3B, C*). However, S1R-4G still formed clusters when reconstituted in GUVs, similar to the wild-type S1R (*Figure 4A, B*). This was in contrast to the diffuse distribution of the S1R-4G mutant in cells (*Figure 3B, C*). To explain these results, we reasoned that the W9L/W11L mutation abolished S1R association with cholesterol, but the 4G mutation only weakened it as this mutation still contains the CARC consensus sequence (*Fantini and Barrantes, 2013*).

To confirm reduced cholesterol binding of the generated S1R mutants, we conducted a series of pulldown experiments with cholesterol agarose (*Palmer et al., 2007*). Cholesteryl hemisuccinate was used for coupling to CarboxyLink reactive resin to generate cholesterol agarose beads. Purified recombinant S1R was incubated with cholesterol agarose, the beads were pelleted by centrifugation, and fractions containing beads and supernatant were analyzed by western blot with anti-S1R antibodies. First, we confirmed that recombinant S1R binds to cholesterol agarose, but not to the control resin (*Figure 4—figure supplement 1*). We then tested S1R-4G and S1R-W9L/W11L mutants. Based on obtained results, we concluded that both mutants have reduced affinity for cholesterol (*Figure 4C, D*). To further confirm these findings, we expressed wild-type and mutant S1R-6His in HEK293 cells, and used cellular lysates from transfected cells in pulldown experiments with cholesterol agarose beads. In these experiments, anti-His tag antibodies were used for detection of recombinant S1R-6His. We found that S1R-W9L/W11L-6His bound to cholesterol agarose significantly weaker than the wild-type S1R (*Figure 4E, F*). In contrast, S1R-4G-6His behaved similar to the wild-type S1R-6His in these experiments (*Figure 4E, F*). Taken together, we concluded that W9L/W11L mutation disrupted cholesterol binding to a much greater extent than 4G mutation. We reason that 4G mutation leads to diffuse distribution of S1R in cells (*Figure 3B*) because ER cholesterol levels in cells are no more than 5 mol % (*Radhakrishnan et al., 2008*). Presumably, these levels of cholesterol are sufficient for association with wild-type S1R but not with S1R-4G mutant that has reduced affinity for cholesterol.

## S1R localizes to, and can generate, thick lipid domains

Our results suggested that S1R is clustered in cholesterol-rich microdomains in the ER (*Figure 3*). It is established that cholesterol-rich phospholipid mixtures have more ordered acyl chains and larger hydrophobic thickness (*de Meyer and Smit, 2009*). Interestingly, according to the crystal structure and hydrophobicity analysis, the length of the S1R transmembrane (TM) domain is 24 a.a (*Figure 3A*; *Schmidt et al., 2016*), which is longer than a typical 20 a.a TM length of ER-resident proteins (*Sharpe et al., 2010*). The length of the S1R TM domain is estimated to be 36.9 Å and should have a positive hydrophobic mismatch when compared to a DOPC bilayer, which has a hydrophobic thickness of 26.8 Å (*Kučerka et al., 2006*). In contrast, a cholesterol-rich DOPC bilayer has a hydrophobic thickness of 36.0 Å, matching well with the S1R TM length (*Milovanovic and Jahn, 2015*). It is therefore possible that S1R clustering in ER membrane is influenced by local changes in membrane thickness, as has been described for the targeting of plasma membrane proteins to lipid rafts (*Lorent et al., 2017*). To test whether the length of the S1R TM also contributes to clustering, we deleted four amino acid residues in its TM region to generate the S1R-Δ4 mutant (*Figure 5A*). When S1R-Δ4 GFP-fusion protein was expressed in HEK293 cells, it displayed a more diffuse distribution in the ER when compared to the wild-type S1R (*Figure 5A*). Mander's coefficient for this mutant is 0.72 ± 0.07 (n = 3, 27 cells) compared to Mander's coefficient of 0.50 ± 0.02 for the wild-type protein (*Figure 3C*). Lack of MAM localization for S1R-Δ4 mutant was confirmed in TOM20 staining experiments (*Figure 5A*).

To test the role of positive hydrophobic mismatch in S1R clustering, we reconstituted purified Alexa647-labeled S1R in GUV membranes composed of 1,2-dinervonoyl-sn-glycero-3-phosphocholine (DNPC) lipids that have a hydrophobic thickness of 35.5 Å. S1R did not form clusters in DNPC GUVs neither in the absence nor in the presence of 20% cholesterol (0%: 6.8%, 92 liposomes, 20%: 2.9%, 77 liposomes) (*Figure 5B*). From these results, we concluded that local increase in membrane thickness contributes to S1R targeting to cholesterol-enriched microdomains in the ER. We then hypothesized that by clustering with cholesterol, S1R can also promote the formation of microdomains with thicker bilayer structure. To measure local membrane thickness in these microdomains, we utilized two designed molecular rulers that report on transmembrane thickness – MBP-TM17, with a short transmembrane helix, and MBP-TM27, with a longer transmembrane helix (*Figure 5C*). The amino acid sequences of TM17 and TM27 were designed based on the synthetic

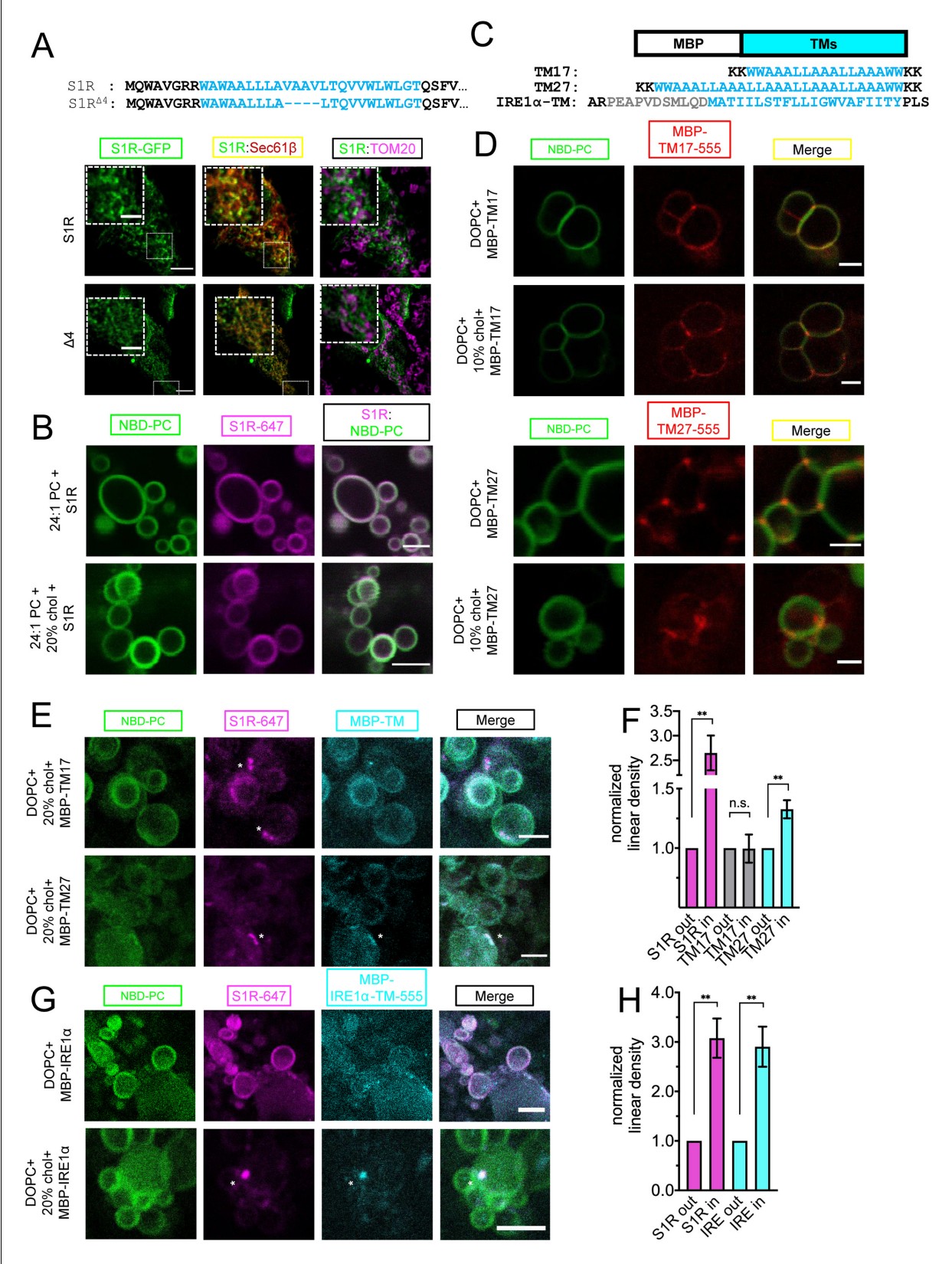

**Figure 5.** Importance of membrane bilayer thickness for S1R cluster formation. (**A**) Deletion of the four amino acid stretch from the S1R transmembrane (TM) domain (top) and intracellular localization of full-length (WT) and shortened mutant (S1R-Δ4) in HEK293 cells. S1R-GFP in green, mCherry-Sec61β in *Figure 5 continued on next page*

*Figure 5 continued*

red, and anti-TOM20 in magenta. Scale bars = 10 µm, insets = 2.5 µm. Mander's colocalization coefficient for the Δ4 mutant in plotted on *Figure 3C*.
(**B**) Distribution of S1R-647 (magenta) in 24:1 phosphatidylcholine (PC) giant unilamellar vesicles (GUVs) (NBD-PC in green) in the absence (top) or presence of 20 mol % cholesterol (bottom). Scale bars = 10 µm. (**C**) Construct design and primary amino acid sequences of MBP-TM17, -TM27, and -IRE1α-TM proteins. Synthetic TM domain in shown in cyan. Construct design of MBT-TMs was based on *Kaiser et al., 2011*. TM helix of IRE1α is shown in cyan and the adjacent amphipathic helix is in gray. Construct design of MBP-IRE1α-TM was based on *Cho et al., 2019*. (**D**) Distribution of purified MBP-TM17-555 (top six panels) and MBP-TM27-555 (lower six panels) in DOPC GUVs in the absence or presence of 10 mol % cholesterol.MBP-TMs are shown in red and NBD-PC in green. Scale bars = 5 µm (MBP-TM17 panels) and 2.5 µm (MBP-TM27 panels). (**E**) Distribution of S1R-647 (magenta) co-reconstituted together with MBP-TM17-555 (cyan, top panel) or MBP-TM27-555 (bottom panel) in DOPC GUVs in the presence of cholesterol (NBD-PC in green). S1R clusters are labeled with asterisks. Scale bars = 5 µm (MBP-TM17) and 2.5 µm (MBP-TM27). (**F**) Linear density of S1R-647 (magenta), MBP-TM17-555 (gray), and MBP-TM27-555 (cyan) outside and inside S1R clusters. Data is mean ± SEM. p-values (n.s. p>0.05, ** p-value<0.01): S1R in vs. S1R out (n = 9): p-value=0.002, TM17 in vs. TM17 out (n = 5): p-value=0.975, TM27 in vs. TM27 out (n = 8): p-value=0.004 based on two-tailed t-test. (**G**) Distribution of S1R-647 (magenta) co-reconstituted together with MBP-IRE1α-TM-555 (cyan) in GUV (NBD-PC in green) in the absence (top) and presence (bottom) of 20 mol % cholesterol. S1R cluster is labeled with an asterisk. Scale bars = 10 µm (top panels) and 5 µm (bottom panels). (**H**) Linear density of S1R-647 (magenta) and MBP-IRE1α-TM-555 (cyan) outside and inside S1R clusters. Data is mean ± SEM. p-values: S1R in vs. S1R out (n = 6): **p-value=0.003, IRE1α in vs. IRE1α out (n = 6): ** p-value=0.005 based on two-tailed t-test.

The online version of this article includes the following figure supplement(s) for figure 5:

**Figure supplement 1.** Purification of MBP-TM17, -TM27 and IRE1α-TM.
**Figure supplement 2.** Distribution of S1R-647 and MBP-TMs in the absence of cholesterol.
**Figure supplement 3.** Impairement of the IRE1α signaling in S1R KO HEK293 cells.

---

peptides previously used in GUV studies of membrane thickness (*Kaiser et al., 2011*). Similar WALP-GFP fusion proteins were successfully used for measurements of the ER lipid heterogeneity in yeast cells (*Prasad et al., 2020*). The hydrophobic length of MBP-TM17, calculated using 1.5 Å rise per residue (*Hildebrand et al., 2004*), matches well with the hydrophobic thickness of the DOPC bilayer (25.5 Å and 26.8 Å, respectively) (*Kučerka et al., 2006*). The calculated hydrophobic length of TM27 is longer (40.5 Å) and matches well to a lipid bilayer with longer acyl chains or a bilayer with high cholesterol content (DOPC+30–40 mol % cholesterol, 36.0 Å) (*Milovanovic et al., 2015*). MBP-TM17 and MBP-TM27 proteins were expressed in *Escherichia coli*, purified using affinity and size exclusion chromatographies, and then covalently labeled with Alexa555 dye (*Figure 5—figure supplement 1*). The MBP-TM17 and MBP-TM27 were first reconstituted into DOPC GUVs in the absence or presence of 10% cholesterol. We found that in the cholesterol-free conditions , MBP-TM17 was distributed largely uniformly in the GUV membrane while MBP-TM27 was enriched at the junctions between the individual GUVs (*Figure 5D*). Cholesterol had little effect on the distribution of MBP-TMs (*Figure 5D*). We hypothesized that MBP-TM27 localized to the GUV junction sites based on a hydrophobic matching mechanism. Interestingly, clustering of S1R at the sites of contacts between different GUVs was occasionally observed in the absence of cholesterol (*Figure 2C*, left panel), most likely due to increase in local membrane thickness at these sites. When S1R and MBP-TM proteins were co-reconstituted together, they were distributed diffusely in the membrane in the absence of cholesterol (*Figure 5—figure supplement 2*). When these rulers were reconstituted together with S1R in the presence of 20% cholesterol, they displayed a different behavior: while MBP-TM17 remained largely diffuse (*Figure 5E*, top panel), MBP-TM27 formed clusters that colocalized with S1R clusters (*Figure 5E*, bottom panel). Analysis of linear density of MBP-TM proteins inside and outside S1R clusters confirmed that MBP-TM27 was enriched in S1R domains, while MBP-TM17 showed no partitioning to S1R clusters (*Figure 5F*). Importantly, this behavior was observed for an artificially designed MBP-TM27 protein, suggesting that recruitment and clustering events were mainly driven by lipid-protein interactions. These results suggested that in the presence of cholesterol S1R induces formation of membrane microdomains with increased local thickness.

To further validate that the same protein recruitment mechanism can be observed for known S1R biological partners, we extended our analysis to an inositol-requiring enzyme 1α (IRE1α). S1R and IRE1α localize in a close proximity to each other in the ER, but do not necessarily interact directly (*Rosen et al., 2019*). In addition to protein-protein interactions, IRE1α oligomerization and stress-response activity can be also modulated by the surrounding lipid environment (*Cho et al., 2019*; *Halbleib et al., 2017*). Loading ER with cholesterol affects IRE1α signaling (*Feng et al., 2003*). First, to confirm colocalization of S1R and IRE1α in cells, we utilized a proximity-labeling approach

(*Hung et al., 2016*). HEK293 cells were transiently transfected with S1R-APEX2 protein or control ER-targeted constructs APEX2-KDEL or Sec61β-APEX2. Following transfection, cells were incubated with biotin-phenol and exposed to $H_2O_2$ for a short period of time to induce biotinylation of proteins in proximity to APEX2. Cell lysates were collected, and biotinylated proteins were pulled down using streptavidin-agarose. The eluate containing biotinylated proteins was analyzed by western blot. Significantly more biotinylated IRE1α was pulled down from the cells expressed S1R-APEX2 than from the cells expressing APEX2-KDEL or Sec61β-APEX2 control constructs (*Figure 5—figure supplement 3*). These results suggested close proximity of IRE1α and S1R-APEX2, consistent with the previous studies (*Mori et al., 2013*; *Rosen et al., 2019*). Short-term (2 hr) stress induction with 1 µM thapsigargin (Tg) had no measurable effect on colocalization of S1R and IRE1α (*Figure 5—figure supplement 3*).

To test whether S1R co-assembles with IRE1α in GUV membranes, we used a similar approach that we previously utilized for the MBP-TM rulers (*Figure 5C*). For these experiments, we generated an expression construct MBP-IRE1α-TM that consisted of the MBP protein fused to the transmembrane helix of human IRE1α and the upstream amphipathic helix (433–464 a.a) (*Figure 5C*; *Cho et al., 2019*). MBP-IRE1α-TM protein was expressed in bacteria, purified, and labeled with Alexa555 (*Figure 5—figure supplement 1*). MBP-IRE1α-TM protein labeled with Alexa555 was co-reconstituted together with S1R labeled with Alexa647 in GUVs and imaged by fluorescence confocal microscopy. In the absence of cholesterol, the distributions of MBP-IRE1α-TM and S1R were uniform (*Figure 5G*, top panel). However, in the presence of 20% cholesterol, MBP-IRE1α-TM was recruited to S1R-positive clusters (*Figure 5G, H*, bottom panel). These results suggested that the transmembrane domain of IRE1α partitions into a lipid microenvironment established by S1R in the presence of cholesterol.

To confirm the functional relevance of IRE1α localization to S1R microdomains in the ER, we compared IRE1α-mediated signaling in wild-type, and S1R knockout (KO) HEK293 cells. S1R KO HEK293 cell line was generated by CRISPR approach, using the same procedure used to generate S1R KO MEF cells in our previous study (*Ryskamp et al., 2017*). Western blotting experiments confirmed absence of S1R in S1R KO HEK293 cells (*Figure 5—figure supplement 3*). Unfolded protein response (UPR) was induced in these cells by addition of 1 µM Tg and levels of IRE1α phosphorylation were examined by western blot using anti-phospho-IRE1α antibody. We determined that under resting conditions the level of phosphorylated IRE1α was lower in S1R KO cells when compared to wild-type cells (*Figure 5—figure supplement 3*). Under ER stress conditions, the same peak level of IRE1α phosphorylation could not be achieved in S1R KO cells (*Figure 5—figure supplement 3*), and total cellular response, measured as integrated area under the curve (*Figure 5—figure supplement 3*), was significantly smaller in S1R KO cells compared to WT, suggesting an impairment in IRE1α signaling. To further examine the activity of IRE1α, we quantified levels of the XBP1s protein, a known downstream effector of IRE1α (*Calfon et al., 2002*). We found that production of XBP1s was delayed in S1R KO cells when compared to wild-type cells (*Figure 5—figure supplement 3*). Taken together, these results suggested that recruitment of IRE1α to S1R-organized microdomains in the ER facilitated IRE1α-mediated signaling. Our conclusions are supported by previously reported inhibition of IRE1α activity in cells transfected with S1R RNAi (*Mori et al., 2013*).

## The dynamic organization of S1R clusters in double supported lipid bilayers

Based on their biological activity, agonists and antagonists of S1R have been described (*Maurice and Su, 2009*). Previous research indicated that S1Rs can redistribute in the ER upon ligand stimulation (*Hayashi and Su, 2003a*; *Hayashi and Su, 2007*). In the next series of experiments, we sought to understand whether S1R ligands can analogously affect the formation and stability of S1R clusters in vitro. It is technically difficult to measure time-resolved dynamics of protein clustering in the membrane of GUVs using standard confocal microscopy. Thus, we used total internal reflection fluorescence (TIRF) microscopy and supported lipid bilayers (SLBs), a system commonly employed to measure the dynamics of membrane-associated proteins (*Ditlev, 2021*; *Su et al., 2016b*) SLBs are typically formed on a glass or mica surface, but in this format transmembrane proteins bind to the underlying glass/mica and usually become immobilized. Recently, several groups reported formation of polyethylene-glycol (PEG)-cushioned bilayers that retained mobility of

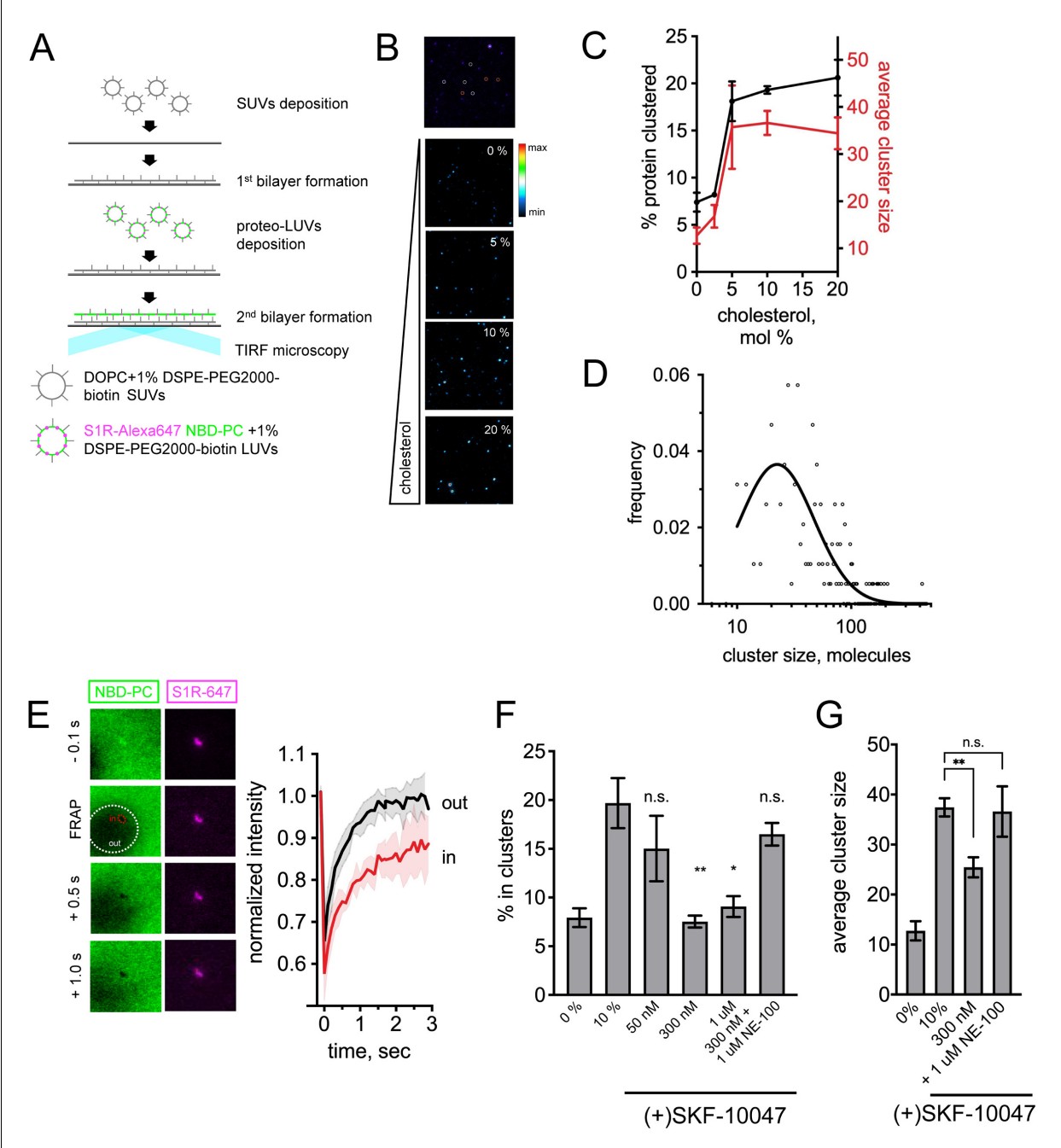

**Figure 6.** Visualization of S1R clusters in double supported lipid bilayers (DLSBs). (**A**) The DSLB technique. Small unilamellar vesicles (SUVs) prepared from DOPC and 1% PEGylated lipids are deposited on a glass surface. S1R-Alexa647 (magenta) is reconstituted into large unilamellar vesicles (LUVs) doped with 1% PEGylated lipids and NBD-labeled phosphatidylcholine (NBD-PC) dye (green). Proteoliposomes are deposited on a pre-formed first bilayer. Bilayer is imaged using total internal reflection fluorescence (TIRF) microscopy. (**B**) Detection of S1R clusters in DSLB (top image). Trimeric receptor (orange circles), bigger assemblies (white circles), and large clusters (magenta) can be identified in DSLBs. Cholesterol-dependent clustering of S1R-647 in DSLB (bottom images). Heat map coloring scheme was used for better visibility. (**C**) Fraction of the clustered receptor(black, left Y-axis) versus membrane cholesterol concentration. In each well, at least 10 fields of view were quantified, and experiments were repeated independently n = 3 times (n = 1 for 2.5%, n = 2 for 5.0%). Average cluster size was calculated (red, right Y-axis) for 10 fields of view at each cholesterol concentration. (**D**) Size distribution of higher-order S1R clusters (more than 10 molecules) in DSLB at 20 mol% cholesterol based on their fluorescent intensity (open circles) and lognormal fit of the data (solid line). (**E**) Lipid dynamics in S1R clusters measured by fluorescence recovery after photobleaching (FRAP). A small area of the bilayer was bleached, and fluorescence recovery of NBD-PC dye was monitored outside (out, white area) and inside (in, red area) of a S1R cluster. Normalized FRAP curves (red within the cluster, black for outside bilayer) are plotted from n = 3 FRAP experiments. (**F**) Ligand effects of S1R agonist and antagonist on S1R clustering in DSLB. Increasing concentrations (50 nM to 1 μM) of a selective S1R agonist (+)-SKF-10047 were added

*Figure 6 continued on next page*

Figure 6 continued

to the wells during the second bilayer formation step (2 hr) and clustered fraction was quantified. S1R antagonist NE-100 at 1 µM concentration was added together with 300 nM (+)-SKF-100047. Data are mean ± SEM. In each experiment, at least 10 fields of view were analyzed. p-values (n.s. p>0.05, *p-value<0.05, **p<0.01): 0% vs. 10%: p=0.005, 50 nM vs. 10%: p=0.974, 300 nM vs. 10%: p=0.003, 1 µM vs. 10%: p=0.136, 300 nM +1 µM NE-100: p=0.985 based on one-way ANOVA with Tukey's post hoc test. (G) Ligand effects of (+)-SKF-10047 on the size of the protein clusters. Average cluster size calculated for at least 10 fields of view for each experimental condition. Data is mean ± SEM. (n.s. p>0.05, **p<0.01): 0% vs. 10%: p=0.0001, 10% vs. 300 nM: p=0.004, 10% vs. 300 nM + 1 µM NE-100: p>0.999 based on one-way ANOVA with Tukey's post hoc test.

The online version of this article includes the following figure supplement(s) for figure 6:

**Figure supplement 1.** Characterization of the double supported lipid bilayer technique.

**Figure supplement 2.** Size distribution of higher-order S1R clusters in double supported lipid bilayer based on their fluorescent intensity at different concentrations of cholesterol (open circles) and lognormal fit of the data (solid lines).

transmembrane proteins (*Pace et al., 2015*; *Richards et al., 2016*; *Wong et al., 2019*). When we attempted to form supported or cushioned bilayers with reconstituted S1R, we observed that protein and lipids were static, presumably due to interaction with the glass surface (data not shown). To overcome these limitations, we modified the bilayer formation procedure to generate double supported lipid bilayers (DSLBs). In DSLBs, PEGylated lipids are used as supports between the two separate bilayers: a lower bilayer that contacts the glass surface is composed of DOPC and 1 mol % DSPE-PEG2000-biotin lipids and an upper bilayer that rests on the first and is composed of lipids and purified transmembrane proteins. To build the DSLB, the first bilayer is formed by fusion and spreading of small unilamellar vesicles (SUVs), prepared by freeze-thaw of DOPC and 1 mol % DSPE-PEG2000-biotin lipids, on a clean glass surface (*Figure 6A*). The second bilayer is formed by deposition of extruded LUVs containing 0.1% NBD-PC as a membrane dye that can be used to measure membrane lipid fluidity by TIRF microscopy (*Figure 6A*). Fluorescence recovery after photobleaching (FRAP) of NBD-PC indicated that lipids in the upper bilayer are highly mobile (*Figure 6—figure supplement 1*). To incorporate protein in the DSLB, purified S1R-Alexa647 was first reconstituted in LUVs using the same approach as for the GUV studies. Protein-loaded LUVs were then applied to the pre-formed lower DOPC bilayer and allowed to spread and form a continuous upper bilayer (*Figure 6A*). The second bilayer was within the TIRF penetration depth (100 nm). FRAP experiments indicated that S1R molecules in this system are mobile and can quickly diffuse laterally in the membrane (*Figure 6—figure supplement 1*).

We discovered that the S1R distribution in DSLBs was heterogeneous, and that small and large clusters could be identified based on their intensity (*Figure 6B*). Small clusters of S1R were highly mobile in the DSLB membrane, whereas large clusters were relatively static. To investigate effects of cholesterol on S1R clustering, we incorporated S1R-647 into DSLBs containing increasing amounts of cholesterol (*Figure 6B*). In agreement with the GUV data (*Figure 2*), we found that increased cholesterol in DSLBs caused S1R to cooperatively assemble into large clusters (*Figure 6B*). The threshold for S1R clustering was in a narrow range between 2.5 and 5.0 mol % (*Figure 6C*, black line), consistent with a typical ER cholesterol content (*Radhakrishnan et al., 2008*). To determine the number of S1R molecules in each of these clusters, we divided the integrated intensity of each cluster by the intensity of a mono-labeled His-pLAT-647 protein molecule (*Su et al., 2016a*). This analysis revealed that small S1R clusters most likely corresponded to S1R trimers, containing on average 2.65 ± 0.58 (n = 47) molecules, consistent with the crystal structure of the protein (*Schmidt et al., 2016*). The number of S1R molecules in the large clusters was widely distributed in a range between tens and hundreds of molecules, with a mean value of 38 molecules per cluster at 20 mol % cholesterol (*Figure 6D*). Titration of cholesterol caused the right-shift in molecular weight distribution (*Figure 6—figure supplement 2*) and also increased the average cluster size (*Figure 6C*, red line). To measure the relative lateral mobility of the lipids, we performed fluorescence recovery after photobleaching (FRAP) experiments on NBD-PC within S1R clusters and in the surrounding membrane (in the presence of 20% cholesterol). Following photobleaching, the recovery of NBD-PC signal was complete outside of S1R clusters ($Y_{max}$(out)=1.01 ± 0.06) but incomplete inside these clusters ($Y_{max}$(in)=0.74 ± 0.07, p=0.008) (*Figure 6E*). Higher immobile fraction indicates that a fraction of molecules was immobilized at the bleached spot due to the fact that they were trapped in rigid domains. These data suggest that lateral mobility of the lipids within S1R clusters is reduced, consistent with known properties of lipid rafts in the plasma membrane (*Sezgin et al., 2017*).

We next evaluated the effects of S1R ligands on S1R clustering. These experiments were performed in the presence of 10% cholesterol in the DSLB. Addition of selective S1R agonist (+)-SKF-10047 at the time of the second bilayer formation reduced the number of large S1R clusters (*Figure 6F*). The effects of (+)-SKF-10047 on S1R clustering were concentration-dependent, with half-maximal effect observed at 50 nM (*Figure 6F*). The effects of (+)-SKF-10047 were blocked by the addition of a S1R antagonist NE-100 (*Figure 6F*). Consistently, agonists resulted in reduced cluster size (*Figure 6G*). These results suggested that S1R agonists prevented formation of S1R clusters or promoted disassembly of S1R clusters in the membrane and that these effects could be blocked by S1R antagonists. Our results are consistent with previous biochemical studies that demonstrated that the proportion of S1R multimers formed in cells was decreased by the agonists (+)-pentazocine and PRE-084 but increased by the antagonists CM304, haloperidol, and NE-100 (*Hong, 2020*; *Hong et al., 2017b*). These results are also consistent with effects of (+)-pentazocine and haloperidol described in FRET experiments performed with cells transfected with fluorescently tagged S1Rs (*Mishra et al., 2015*). Similar conclusions were made when using receptor bioluminescence resonance energy transfer (BRET) homomer assay and non-denaturing gel electrophoresis (*Yano et al., 2018*). In contrast, it has been reported that S1R agonists such as (+)-pentazocine and PRE-084 stabilized the oligomeric state of purified S1R in the presence of detergents (*Gromek et al., 2014*), suggesting that biochemical properties of S1R in the membrane and in detergent may differ from each other. Another potential reason for this discrepancy is that the previous study (*Gromek et al., 2014*) was focused on the analysis of smaller tetrameric species, while our analysis measured larger assemblies of at least ten S1R molecules. Taken together, our results indicate that S1R participates in the formation of thicker cholesterol-rich lipid clusters and S1R agonists act by disassembling these membrane microdomains.

## Discussion

S1R modulates many physiological processes, such as cell excitability, transcriptional activity, $Ca^{2+}$ homeostasis, stress response, and autophagy (*Christ et al., 2019*; *Couly et al., 2020*; *Hayashi, 2019*; *Kourrich, 2017*; *Maurice and Goguadze, 2017*; *Ryskamp et al., 2019*). However, S1R-mediated signal transduction differs from a canonical second messenger-coupled transmembrane receptor signaling, and S1R is often referred to as a 'ligand-gated molecular chaperone' (*Hayashi, 2019*; *Nguyen et al., 2017*; *Su et al., 2010*). In the seminal paper by *Hayashi and Su, 2007*, S1R was proposed to have chaperone-like properties at MAMs. However, the molecular mechanism of chaperone activity of S1R remains unexplained because S1R lacks structural similarity with known chaperones or extensive protein interaction interfaces. Based on our results, we propose that the biological activity of S1R in cells can be explained by the ability of this protein to form cholesterol-enriched microdomains in the ER, therefore acting as ER lipid scaffolding protein. A role of S1R in organization and remodeling of lipid raft microdomains in the plasma membrane has been proposed previously based on cell biological studies (*Palmer et al., 2007*; *Vollrath et al., 2014*). Our data further suggest that such microdomains have increased local membrane thickness, providing a favorable environment for recruitment of ER proteins with longer transmembrane domains (*Lorent et al., 2017*). Increased local cholesterol concentration and membrane thickness can modulate activity of ER proteins recruited to these microdomains. Our hypothesis may explain how a small protein such as S1R is able to modulate activity of almost a hundred effector proteins (*Couly et al., 2020*; *Delprat et al., 2020*; *Kourrich et al., 2012*; *Ryskamp et al., 2019*; *Schmidt and Kruse, 2019*). We reason that activity of these proteins could be affected by changes in local lipid microenvironment and not only via protein-protein interactions with the S1R.

By performing experiments using reduced reconstitution systems, we have been able to demonstrate direct effects of cholesterol on S1R clustering (*Figures 2* and *5*). Previous studies identified two potential sites of S1R association with cholesterol – Y173 and Y201/Y206 (*Palmer et al., 2007*). Our studies suggest that Y201 and Y206 residues are dispensable for cholesterol-mediated S1R clustering (*Figure 3*). Moreover, we identified a tandem CARC-like motif (*Fantini and Barrantes, 2013*) within the transmembrane region of S1R (*Figure 3A*). Mutations of this motif impaired association of recombinant S1R with cholesterol beads (*Figure 4*), affected cholesterol-dependent S1R clustering in GUV reconstitution experiments (*Figure 4*), and disrupted S1R targeting to MAMs and ER-PM contact sites in cells (*Figure 3*). The threshold for formation of S1R clusters was in a narrow range

between 2.5 and 5.0 mol% of cholesterol (*Figure 6C*). Similar cholesterol dependence was previously described for SREBP-2 activation, with half-maximal response at 4.5 mol % of ER cholesterol (*Radhakrishnan et al., 2008*), indicating that S1R affinity for cholesterol is within the physiological range of ER cholesterol concentrations.

A recent study demonstrated that micrometer-sized large intracellular vesicles exhibit phase-separation behavior at contact sites between the ER and mitochondria, plasma membrane and lipid droplets (*King et al., 2020*), similar to phase separation observed in giant plasma membrane-derived vesicles (*Leventhal et al., 2011*). It has been shown that MAMs are enriched in cholesterol and ceramides (*Area-Gomez et al., 2012*; *Hayashi and Fujimoto, 2010*; *Hayashi and Su, 2003a*). Our findings suggest that S1R can contribute to stabilization and/or formation of these cholesterol-rich lipid microdomains in the ER membrane. This conclusion is consistent with MAM defects observed in S1R KO mice (*Watanabe et al., 2016*), with earlier analysis of S1R targeting to detergent-resistant domains in the ER (*Hayashi and Fujimoto, 2010*), and with previous suggestions that S1R contributes to stability of lipid rafts in the plasma membrane (*Palmer et al., 2007*; *Vollrath et al., 2014*).

The lipid regulation of PM proteins is a well-known phenomenon (*Rosenhouse-Dantsker et al., 2012*). It was shown in direct and indirect experiments that activities of several ER channels including ryanodine receptors, sarco/endoplasmic reticulum $Ca^{2+}$-ATPase, and inositol-1,4,5-triphosphate receptors can be modulated by cholesterol content, lipid packing, or membrane thickness (*Cannon et al., 2003*; *Gustavsson et al., 2011*; *Li et al., 2004*; *Madden et al., 1979*; *Sano et al., 2009*). Components of the gamma secretase complex reside at MAMs, and the transmembrane region of amyloid precursor protein (APP) contains cholesterol-binding motifs (*Montesinos et al., 2020*; *Pera et al., 2017*), suggesting modulatory effects of cholesterol on APP processing. Several ER stress sensors, including IRE1α, can sense membrane saturation and be activated without protein unfolding in the ER lumen (*Ballweg et al., 2020*; *Cho et al., 2019*; *Halbleib et al., 2017*). In our GUV reconstitution experiments, we demonstrated that a minimal sensor derived from IRE1α transmembrane domain is recruited to S1R clusters (*Figure 5G, H*), and in agreement with the previous report (*Mori et al., 2013*), S1R cellular KO impaired IRE1α-mediated signaling (*Figure 5—figure supplement 3*).

S1R can exist in an oligomeric form (*Gromek et al., 2014*; *Mishra et al., 2015*). Originally, a photoaffinity probe labeled high-molecular weight oligomers in rat liver microsomes (*Pal et al., 2007*). High-molecular weight species up to 400 kDa were later detected in membrane preparations using various biochemical approaches (*Yano et al., 2018*; *Yano et al., 2019*), and a similar wide molecular weight distribution was observed for the recombinant protein (*Schmidt et al., 2016*). The functional relevance of the S1R oligomerization is, however, not clear. In our experiments, we observed that S1R oligomers form continuous molecular weight distribution ranging from trimers to large (10–100 subunits) clusters (*Figure 6D*). Formation of large clusters was facilitated by the presence of cholesterol. S1R agonists disrupted large S1R clusters in cholesterol-containing lipid bilayers in a concentration-dependent manner (*Figure 6F, G*). Our results are consistent with ligand effects observed by native electrophoresis from cell membranes and S1R-BiP association (*Hayashi and Su, 2007*; *Hong, 2020*; *Hong et al., 2017b*). Also in agreement with our findings, S1R agonist (+)-SKF-10047 resulted in destabilization of lipid rafts in the plasma membrane of cancer cells (*Palmer et al., 2007*).

Mutations in S1R lead to a juvenile form of ALS (*Al-Saif et al., 2011*) and distal hereditary motor neuropathies (*Almendra et al., 2018*; *Gregianin et al., 2016*; *Horga et al., 2016*; *Li et al., 2015*; *Ververis et al., 2020*). We demonstrated that an ALS-causing E102Q variant acted as a loss-of-function mutant that disrupted S1R targeting to MAMs (*Figure 1—figure supplement 1*). However, E102 residue is located outside of cholesterol-binding region of S1R, and this mutation likely acts by a different mechanism, such as causing S1R misfolding, aggregation, or a change in oligomerization state (*Abramyan et al., 2020*). S1R is considered to be a potential drug target for treatment of neurodegenerative disorders and cancer (*Herrando-Grabulosa et al., 2021*; *Kim and Maher, 2017*; *Maurice and Goguadze, 2017*; *Maurice and Su, 2009*; *Nguyen et al., 2017*; *Ryskamp et al., 2019*). S1R agonists demonstrated neuroprotective effects in a variety of neurodegenerative disease models and are currently in clinical trials for a variety of neurological disorders including Alzheimer's, Huntington's, Parkinson's diseases and ALS (*Brimson et al., 2020*; *Herrando-Grabulosa et al., 2021*; *Maurice and Goguadze, 2017*; *European Huntington's Disease Network et al., 2019*;

*Ryskamp et al., 2019*). It has been proposed that accumulation of C99 fragment of APP leads to enhanced formation of cholesterol microdomains in the ER and upregulation of MAM activity in AD neurons (*Area-Gomez et al., 2012*; *Montesinos et al., 2020*; *Pera et al., 2017*). In contrast, MAM downregulation was observed in motor neurons in a genetic model of ALS (*Watanabe et al., 2016*). Our results suggest that S1R agonists allow remodeling of lipid microdomains in the ER membrane, which may help to normalize MAM function in AD and ALS neurons. Long-term effects of S1R activation may include a more thorough remodeling of MAMs including size and protein composition needed for long-lasting metabolic adjustments in stress conditions. Changes in ER lipid microenvironment may also affect function of a variety of channels, transporters, and other signaling proteins localized to S1R signaling microdomains. The exact functional outcome of these lipid perturbations will be unique to each particular protein based on its activity in a lipid environment established by S1R.

In conclusion, we propose that many biological functions of S1R can be explained by its ability to organize and remodel cholesterol-enriched ER microdomains, which in turn affects activity of ER signaling proteins, stress-response, and metabolic status of the cells.

## Materials and methods

Key resources table is included in supplementary materials.

Raw data and tables used for generation of the figures are available at https://doi.org/10.5061/dryad.9zw3r22dn.

### Cell culture and transfection

HEK293T cells were cultured in DMEM medium supplemented with 10% fetal bovine serum. Transfection was performed using Lipofectamine LTX Plus according to the manufacturer's recommendations. Cells were routinely tested for mycoplasma contamination using DAPI staining and PCR mycoplasma detection kit (MD Biosciences).

### Lipids and detergents

*Lipids*: 1,2-dioleoyl-sn-glycero-3-phosphocholine (18:1 PC, DOPC), 1,2-dinervonoyl-sn-glycero-3-phosphocholine (24:1 PC, DNPC), 1-palmitoyl-2-{6-[(7-nitro-2–1,3-benzoxadiazol-4-yl)amino]hexanoyl}-sn-glycero-3-phosphocholine (NBD-PC), cholesterol, 1,2-distearoyl-sn-glycero-3-phosphoethanolamine-N-[biotinyl(polyethylene glycol)−2000] (DSPE-PEG(2000)-biotin), 1,2-dioleoyl-sn-glycero-3-[(N-(5-amino-1-carboxypentyl)iminodiacetic acid)succinyl] (nickel salt, DGS-NTA-Ni), L-α-phosphatidylcholine from chicken egg (Egg PC) were purchased from Avanti Polar Lipids.

*Detergents*: n-dodecyl-N,N-dimethylamine-N-oxide (LDAO), n-dodecyl-β-D-maltopyranoside (DDM), n-octyl-β-D-glucopyranoside (OG), and cholesteryl hemisuccinate (CHS) were purchased from Anatrace.

### Expression plasmids

Plasmid encoding human S1R gene fused with GFP (S1R-GFP) was generated by PCR amplification of the human S1R gene (http://www.ncbi.nlm.nih.gov/nuccore/NM_005866.3) and cloning into pEGFP-N2 vector (Clontech) using HindIII/XbaI cloning sites. mCherry-Sec61β was obtained from Addgene (https://www.addgene.org/49155) (*Zurek et al., 2011*). Mutations were introduced using Q5 site-directed mutagenesis kit (NEB) according to the manufacturer's instructions. mCherry-MAPPER plasmid was generated by replacing GFP with mCherry in the original construct described in *Chang et al., 2013*. For baculovirus expression of 6His-tagged S1R, human S1R gene was amplified by PCR and cloned into pFastBac-HTA vector (Bac-to-Bac baculovirus expression system, Thermo Fisher Scientific) using EcoRI/HindIII sites. Mutations were introduced using Q5 site-directed mutagenesis kit (NEB). For expression of S1R-6His mutants in HEK293 cells, genes were amplified from S1R-GFP constructs and cloned into lentivector expression plasmid (FUGW, addgene.org/14883) using XhoI/BamHI cloning sites. The reverse primer contained a 6His encoding sequence. Nucleic acid sequences encoding TM17, TM27, and human IRE1α-TM were synthesized by GenScript. Genes corresponding to the transmembrane peptides with following amino acid sequences, TM17 KK WWAAALLAAALLAAAWWKK and TM27 KKWWAAALLAAALLAAALLAAAWWKK, were synthesized by GenScript and cloned into pMAL-c5x vector using XmnI/NotI sites. A gene encoding a

fragment of human IRE1α (a.a.r. 431–467 of the transmembrane helix and adjacent amphipathic helix), corresponding to the following amino acid sequence, ARPEAPVDSMLQDMATIILSTFLLIG WVAFIITYPLSK with a single point mutation K441Q, was synthesized by GenScript and cloned into pMAL-c5x using XmnI/NotI restriction sites. Genetic sequences of IRE1α, TM17, and TM27 were codon optimized for *E. coli* expression by GenScript. For cloning S1R-APEX2 fusion gene, APEX2 (https://www.addgene.org/49386) and human S1R genes were amplified by PCR. NotI site was introduced to the APEX2 5′ primer and to the S1R 3′′ primer. PCR product was ligated using T4 ligase (NEB) and amplified using outer primers to produce the fusion gene, S1R-APEX2. Resulting APEX2-S1R gene was cloned into lentivector expression plasmid (FUGW, addgene.org/14883). Control plasmid encoding Sec61β-APEX2 (https://www.addgene.org/83411) was obtained from Addgene (*Lee et al., 2016*), and APEX2-KDEL was generated by adding KDEL-encoding gene sequence to the reverse primer. Sec61β-APEX2 and APEX2-KDEL fusion genes were cloned into lentivector plasmid FUGW using XbaI/EcoRI sites. All constructs were sequenced to confirm the accuracy of cloning.

## Protein expression, purification, and NHS-conjugated dye labeling

### Purification of S1R

S1R was expressed with an N-terminal 6His-tag fusion in Sf9 cells using Bac-to-Bac baculoviral expression system (Thermo Fisher Scientific) according to the manufacturer's recommendations. Infection was performed at cell density of $2 \times 10^6$ cells/ mL, cells were collected 68 hr post-infection, and cell pellet was stored at −80°C.

Cells were lysed in hypotonic buffer (20 mM HEPES pH = 8.0, 1x cOmplete EDTA-free protease inhibitor cocktail [Roche]) and sonicated three times for 2 min. Cell debris was centrifuged at 6000 *g* for 15 min at 4°C. Supernatant was collected and pellet sonication was repeated one more time. After the second centrifugation step, supernatants were combined together and centrifuged at 100,000 *g* for 1 hr in a Ti70 rotor (Beckman Coulter). Membrane pellet was resuspended in a solubilization buffer (50 mM HEPES pH = 8.0, 300 mM NaCl, 1% LDAO, 0.1% cholesteryl hemisuccinate) using glass tissue homogenizer and rotated overnight at 4°C. Next day, solubilized membranes were centrifuged at 100,000 *g* for 1 hr. Supernatant was incubated with 1 mL of $Ni^{2+}$-NTA agarose (Invitrogen) for 1 hr and washed with 25 mL of 50 mM HEPES pH = 8.0, 300 mM NaCl, 1% LDAO, 0.1% cholesteryl hemisuccinate, 10 mM imidazole, and then with the same volume of buffer containing 20 mM imidazole. The protein was eluted in the same buffer containing 150 mM imidazole.

Receptor was further purified using anion-exchanged chromatography using HiTrap Q HP column (GE Healthcare). Protein was diluted 1:10 with buffer A (20 mM MOPS pH = 7.0, 0.1% LDAO, 0.01% cholesteryl hemisuccinate, 0.0015% Egg PC) and loaded onto a column, washed with five volumes of buffer A and eluted using linear gradient of buffer B (20 mM MOPS pH = 7.0, 1 M NaCl, 0.1% LDAO, 0.01% cholesteryl hemisuccinate, 0.0015% Egg PC). Fractions containing S1R were collected, pooled together, and concentrated using Amicon centrifugal filters with 50,000 Da MWCO (Millipore).

Solution pH was adjusted to pH = 8.0 with HEPES buffer, and protein solution was mixed with an excess of Alexa647 NHS ester (Molecular probes) dissolved in DMSO and left for labeling overnight at 4°C.

Next day, labeled protein was purified using gel-filtration chromatography on Superdex 200 10/ 300 column (GE Healthcare) in buffer containing 50 mM HEPES pH = 8.0, 300 mM NaCl, 0.1% LDAO, 0.01% cholesteryl hemisuccinate, 0.0015% Egg PC. Following the size-exclusion chromatography step, receptor was concentrated to 1–2 mg/mL, flash-frozen in small aliquots in liquid nitrogen, and stored at −80°C.

### Purification of MBP-TM17, -TM27, and -IRE1α-TM

MBP-TM17, -TM27, and -IRE1α-TM were expressed in *E. coli* BL21(DE3) cells. 10 mL of LB medium were inoculated and grown overnight. Next day, the starter culture was added to 1 L of LB medium and grown until OD(600)=0.6. Protein synthesis was induced by addition of 0.3 mM IPTG. Proteins were expressed at 37°C for 3 hr. Cells were collected by centrifugation, and cell pellet was stored at −80°C.

Proteins were purified according to the procedure described in *Halbleib et al., 2017*. Cell pellet was resuspended in lysis buffer (50 mM HEPES pH = 7.4, 150 mM NaCl, 1 mM EDTA, 2 mM DTT, and 1x cOmplete protease inhibitor cocktail [Roche]). Cells were sonicated three times for 2 min each. 500 mM OG stock was added to the final concentration of 50 mM, and lysate was incubated for 10 min at 4°C. Lysate was clarified by centrifugation at 55,000 *g* for 1 hr in 25.50 rotor (Beckman Coulter). Clarified supernatant was loaded on 1.5 mL of amylose resin (NEB) and washed with 50 mL of 50 mM HEPES pH = 7.4, 150 mM NaCl, 1 mM EDTA, 2 mM DTT, 50 mM OG, and then with 50 mL of buffer without DTT. Protein was eluted in the same buffer containing 10 mM maltose.

After concentration with Amicon centrifugal filters with 50,000 Da MWCO (Millipore), protein was labeled with Alexa555 NHS ester (Molecular Probes) as described for S1R. Protein was purified using Superdex 200 10/300 column (GE Healthcare) in a buffer containing 50 mM HEPES pH = 7.4, 150 mM NaCl, 1 mM EDTA, 0.018% DDM, concentrated, aliquoted, flash-frozen in liquid nitrogen, and stored at −80°C.

## Purification of LAT

LAT was purified as described in *Su et al., 2016a*. BL21(DE3) cells containing MBP-8*His-LAT 48-233-6His were collected by centrifugation and lysed by cell disruption (Emulsiflex-C5, Avestin) in 20 mM imidazole pH = 8.0, 150 mM NaCl, 5 mM βME, 0.1% NP-40, 10% glycerol, 1 mM PMSF, 1 μg/mL antipain, 1 μg/mL pepstatin, and 1 μg/mL leupeptin. Centrifugation-cleared lysate was applied to $Ni^{2+}$-NTA agarose (Qiagen), washed with 10 mM imidazole pH = 8.0, 150 mM NaCl, 5 mM βME, 0.01% NP-40, and 10% glycerol, and eluted with the same buffer containing 500 mM imidazole pH = 8.0. The MBP tag and 6His-tag were removed using TEV protease treatment for 16 hr at 4°C. Cleaved protein was applied to a Source 15 Q anion exchange column and eluted with a gradient of 200–300 mM NaCl in 20 mM HEPES pH = 7.0 and 2 mM DTT followed by size-exclusion chromatography using a Superdex 200 prepgrade column (GE Healthcare) in 25 mM HEPES pH = 7.5, 150 mM NaCl, 1 mM $MgCl_2$, and 1 mM DTT. LAT was exchanged into buffer containing no reducing agent (25 mM HEPES pH = 7.0, 150 mM NaCl, 1 mM EDTA) using a HiTrap Desalting Column. $C_2$-maleimide Alexa647 were added in excess and incubated with protein for 16 hr at 4°C or 2 hr at room temperature. Following the incubation, 5 mM βME was added to the mixture to quench the labeling reaction. Excess dye was removed from labeled protein by size-exclusion chromatography.

## Liposomal preparation and protein insertion for GUV experiments

### Reconstitution to liposomes

For protein reconstitution to liposomes, dry lipid films were prepared by dissolving lipids in chloroform with 0.1 mol % NBD-PC membrane label dye and drying overnight under vacuum. Next day, films were rehydrated in 20 mM HEPES pH = 8.0 buffer to the final concentration of 2 mg/mL and extruded using liposomal mini-extruder (Avanti Polar Lipids) through 0.1 μm pore size polycarbonate filters (Avanti Polar Lipids). Liposomes were mixed with DDM (final 0.8 mM) and purified S1R at a lipid-to-protein ratio of 200:1. Mixture was incubated at room temperature for 1 hr. Detergent was removed by three additions of BioBeads SM-2 adsorbent resin (25 mg per 1 mL of lipid mixture) (BioRad) for 2 hr each at 4°C. For DNPC reconstitution, initial incubation and BioBeads adsorption was performed at 27°C. Prepared liposomes were used on the day of experiment.

For co-reconstitution experiments with MBP-TM17, -TM27, and -IRE1α-TM, purified MBP fusion proteins were added at the initial incubation step at equimolar concentration.

### Preparation of GUVs

GUVs were formed using the polymer-assisted swelling on an agarose gel (*Horger et al., 2009*). Glass slides were covered with 1.0% agarose and dried to completion on a hot plate. Small imaging chambers were assembled using adhesive silicon insulators (Electron Microscopy Sciences). Sucrose was added to the proteoliposomes at the final concentration of 15 mM, and 0.3 μL drops were deposited on the agarose-coated coverslips. After dehydration for 10 min at room temperature, slides were rehydrated in 50 mM HEPES pH = 8.0, 150 mM NaCl buffer, and incubated on a hot plate for 1 hr at 42°C. Samples were prepared side by side and imaged within 15–20 min after GUV formation. GUVs were imaged using upright fluorescent confocal microscope (Leica) with a 63×

water immersion objective. In cluster counting experiments, cluster-positive liposomes were quantified by a blinded independent observer.

## Preparation of DSLBs

For SUV preparation, lipid films of DOPC and 1.0 mol % DSPE-PEG(2000)-biotin were prepared and dried overnight under vacuum. Lipids were rehydrated in a bilayer buffer (50 mM Tris, pH = 7.4, 150 mM NaCl, 1 mM TCEP) to the final concentration of 2.0 mg/mL and freeze-thawed 10 times in liquid nitrogen. Liposomes were stored at −80°C. Prior to experiment, liposomes were centrifuged at 100,000 $g$ for 1 hr using a Sw55 rotor (Beckman Coulter), supernatant was collected, and stored at 4°C up to 2 weeks under argon.

For the second bilayer mixtures, lipid films of DOPC, 1.0 mol% DSPE-PEG(2000)-biotin, 0.1% NBD-PC, and cholesterol at indicated molar concentrations were prepared and dried overnight under vacuum. Lipids were rehydrated to the final concentration of 2 mg/mL and extruded through the 0.1 μm polycarbonate filter as described above. Liposomes were mixed with 0.8 mM DDM and purified S1R at lipid-to-protein ratio of 2000:1. Mixtures were incubated at room temperature for 1 hr, and detergent was subsequently removed by three 2-hr-long additions of BioBeads SM-2 (25 mg per 1 mL of lipid mixture). Proteoliposomes were flash-frozen in liquid nitrogen and stored at −80°C. Prior to experiment, proteoliposomes were extruded through the 0.1 μm polycarbonate filter and centrifuged at 14,000 $g$ for 10 min using a tabletop centrifuge.

Supported bilayers were formed in 96-well plates (MatriPlace MGB096-1-2LG-L, Brooks Life Science Systems). Glass surface was cleaned by immersing the plate in 5% Hellmanex III solution (Hellma Analytics) at 60°C for 3 hr. Plate was thoroughly washed with milliQ water to remove any remaining Hellmanex solution. Wells were dried with argon and sealed with foil tape. On the day of the experiment, wells were cut open and hydrated with 500 μL milliQ water. 300 μL of 6 N NaOH were added to each well, and plate was incubated on a heater for 1 hr at 42°C. Then sodium hydroxide was removed and replaced with 300 μL of fresh solution and incubated for 1 hr. After that, each well was washed three times with 700 μL of milliQ water and three times with 700 μL of bilayer buffer solution (50 mM Tris pH = 7.4, 150 mM NaCl, 1 mM TCEP).

Each well was filled with 200 μL of the bilayer buffer, and 20 μL of SUVs were added. Plate was incubated at 42°C for 2–3 hr, and then each well was washed with 500 μL of the bilayer buffer. After that, proteoliposomes were added to each well (20 μL) and the plate was incubated at 42°C for 2 hr. Where indicated, ligands were added to the bilayer buffer during the second incubation step. Each well was washed 10 times with 500 μL of the bilayer buffer.

For LAT-647 calibration experiments, bilayers were formed as described in *Su et al., 2016a*. Glass plates were cleaned as described above, washed, and incubated with SUVs prepared by freeze-thaw method from 99% DOPC, 1% DGS-NTA-Ni, and 0.1% NBD-PC. Bilayers were formed as described above. After washing three times with bilayer buffer, bilayer was blocked with 1 mg/mL BSA in bilayer buffer for 30 min at room temperature, washed three times and incubated with His-tagged LAT (final 0.1–1.0 pM) for 30 min, and then washed three times to remove unbound protein. TIRF images were captured using a Nikon Eclipse Ti microscope base equipped with an AndoriXon Ultra 897 EM-CCD camera with a 100 × 1.49 NA objective, a TIRF/iLAS2 TIRF/FRAP module (Biovision) mounted on a Leica DMI6000 microscope base equipped with a Hamamatsu ImagEMX2 EM-CCD camera with a 100 × 1.49 NA objective, or a Nikon Eclipse Ti microscope base equipped with a Hamamatsu ORCA Flash 4.0 camera with a 100 × 1.49 NA objective.

## Pulldown experiments

For synthesis of cholesterol-coupled agarose, one resin equivalent of CarboxyLink coupling agarose (Thermo Fisher Scientific) was washed three times in 10 mL of dimethyl formamide (DMF) and mixed with two resin equivalents of cholesteryl hemisuccinate (Avanti Polar Lipids), two resin equivalents of HBTU, and four resin equivalents of DIPEA. For the synthesis of a control resin, acetic acid was used instead of the cholesteryl hemisuccinate. Reaction proceeded for 2 hr at room temperature with constant shaking. Then resin was washed three times in 10 mL of DMF, three times with N,N′-diisopropylcarbodiimide (DIC), three times with methanol, and three times with water. Resin was stored in 20% ethanol.

For pulldown experiments with recombinant proteins, 30 µL of the resin were mixed with 0.2 µg of recombinant His-S1R in 500 µL of binding buffer (20 mM Tris-HCl pH = 8.0, 300 mM NaCl, 0.5 mM EDTA, 0.1% LDAO). Resin was incubated on a shaker for 3 hr at 4°C, washed three times in 500 µL of binding buffer, and proteins were eluted by boiling resin in 1× SDS Laemmli loading buffer.

For pulldown experiments with exogenously expressed S1R, HEK293 cells were transfected with S1R-6His proteins. 48 hr post-transfection, cells were lysed in 50 mM Tris-HCl pH = 8.0, 150 mM NaCl, 1% Triton X-100, 1% LDAO buffer. For pre-clearing step, 10 µg of total cell lysate from non-transfected cells were mixed with 30 µL of resin in 500 µL of binding buffer and incubated at 4°C for 1 hr. Resin was washed twice and incubated with 10 µg of total cell lysate from S1R-6His-expressing cells for 3 hr. Resin was washed thrice, and proteins were eluted by boiling in 1× SDS Laemmli buffer.

## Isolation of MAMs

MAMs were isolated from C57/B6 mouse strain liver by following previously reported procedures (*Wieckowski et al., 2009*). Protein concentration in isolated fractions was measured by Bradford assay. Proteins (10 µg per lane) were separated by SDS-PAGE and analyzed by western blot analysis using organelle-specific antibodies.

## Cell imaging

For imaging S1R localization in HEK293T cells, cells were cultured on glass coverslips in 24-well plates. Each well was transfected with 150 ng of S1R-GFP plasmid (WT or mutant) and 150 ng of mCherry-Sec61β using Lipofectamine LTX Plus (Thermo Fisher Scientific) according to the manufacturer's instructions. Cells were fixed 48 hr post-transfection in 4% paraformaldehyde (PFA) in phosphate buffered saline (PBS) solution for 20 min, permeabilized and blocked with 5% bovine serum albumin (BSA), 0.1% Triton X-100 in PBS for 1 hr, and stained with anti-mCherry (16D7, 1:500, Invitrogen) and anti-TOM20 antibodies (FL-145, 1:500, Santa Cruz Biotechnology) overnight at 4°C. Next day, cells were washed and incubated with secondary antibodies (594 donkey anti-rat, A21209, 1:1000, Invitrogen; 647 goat anti-rabbit, A27040, 1:1000, Invitrogen) for 1 hr at room temperature. Cells were washed thrice with PBS and mounted using Aqua Polymount solution (Polysciences). Cells were visualized using fluorescent confocal microscope (Leica) with 63× oil immersion objective.

## TIRF microscopy imaging

HeLa cells were plated on 8-well Lab-Tek chambered coverglass (Nunc) at a density of $1.5 \times 10^4$ cells/well the day before transfection. Plasmid DNA were transfected into cells using TransIT-LT1 (Mirus Bio) with the 1:3 DNA-to reagent ratio. The plasmid DNA used in the transfection are mCherry-MAPPER (50 ng/well) and S1R-GFP (50 ng/well). Cells were washed with extracellular buffer (ECB; 125 mM NaCl, 5 mM KCl, 1.5 mM MgCl₂, 20 mM Hepes, 10 mM glucose, and 1.5 mM CaCl₂, pH = 7.4) before imaging and imaged in the ECB. TIRF microscopy imaging experiments were performed at room temperature with a CFI Apo TIRF 100×/1.49 objective on a spinning-disc confocal TIRF microscope custom-built based on a Nikon Eclipse Ti-E inverted microscope (Nikon Instruments) with a HQ2 camera. The microscope was controlled by Micro-Manager software.

## Expansion microscopy

To prepare expanded specimens, we used a procedure developed by Tillberg (*Tillberg et al., 2016*). Briefly, HEK293T cells were cultured, transfected with S1R-GFP and mCherry-Sec61β plasmids, and stained with primary (anti-GFP, ab13970, 1:500, Abcam; anti-mCherry, 16D7, 1:500, Invitrogen, and anti-TOM20, FL-145, 1:500, Santa Cruz Biotechnology) and secondary antibodies (488 goat anti-chicken, A11039, 1:1,000, Invitrogen; 594 donkey anti-rat, A21209, 1:1000, and Atto647N goat anti-rabbit 40839, 1:1,000, Sigma-Aldrich) as described above. Cells were processed according to protein-retention expansion microscopy (*Tillberg et al., 2016*). Briefly, permeabilized cells were incubated with 0.1 mg/mL 6-((acryloyl)amino)hexanoic acid, succinimidyl ester (AcX, Thermo Fisher Scientific) in PBS overnight, washed three times with PBS, and incubated in a gelation solution (1× PBS, 2 M NaCl, 8.625% [w/w] sodium acrylate, 2.5% [w/w] acrylamide, 0.15% [w/w] N,N'-methylene-bisacrylamide, 0.02% TEMED, 0.02% ammonium persulfate). Samples were transferred to 37°C tissue

culture incubator for 2 hr. After gelation, samples were digested with proteinase K (NEB) diluted to 8 u/mL in digestion buffer (50 mM Tris pH = 8.0, 1 mM EDTA, 0.5% Triton X-100, 1 M NaCl) overnight at room temperature. To enhance the fluorescent signal, samples were washed with PBS, blocked, and restained with primary and secondary antibodies. Then, samples were transferred to milliQ water and allowed to complete expansion (about 1 hr with three water changes). Gel-embedded samples were mounted on poly-L-lysine-coated glass coverslips and visualized using fluorescent confocal microscope.

## CRISPR/Cas9-mediated deletion of S1R

To delete endogenous S1R in HEK293T cells, we used the CRISPR/Cas9 system. GuideRNA sequences targeting mouse S1R were designed using bioinformatics tools (crispr.mit.edu for maximizing specificity and http://www.broadinstitute.org/rnai/public/analysis-tools/sgrna-design for selecting guide sequences with predicted efficacy), and sgRNA plasmids targeting S1R (gS1R) were generated. A sgRNA sequence targeting exon 1 of S1R (GCGCGAAGAGATAGCGCAGT) was subcloned into the lentiGuide-Puro plasmid (addgene.org/52963/) as in *Sanjana et al., 2014* following their protocol (addgene.org/static/data/plasmids/52/52963/52963-attachment_IPB7ZL_hJcbm.pdf). The lenti-Cas9-Blast plasmid (addgene.org/52962/) was used to express Cas9. To validate these plasmids, HEK293 cells were co-transfected with Cas9-Blast and gS1R-Puro plasmids using FuGENE6 (1:4 DNA to charge ratio), and cells transfected with both plasmids were selected with 5 µg/mL blasticidin and 10 µg/mL puromycin in the culture media. Western blotting analysis confirmed efficient deletion of S1R in HEK293 cells.

## APEX2 pulldown assay

For APEX2-based proximity-labeling experiments, we followed a procedure described in *Hung et al., 2016*. Briefly, HEK293T cells cultured on 10 cm$^2$ dishes were transfected with 10 µg of S1R-APEX2, Sec61β-APEX2, or APEX2-KDEL plasmids. 48 hr post-transfection, cells were incubated in 500 µM biotin-phenol (Iris-Biotech) in complete medium at 37°C for 1 hr. Then, proteins were labeled by addition of 1 mM H$_2$O$_2$ for 1 min and quenched with 10 mM sodium ascorbate, 5 mM Trolox, 10 mM sodium azide in PBS. Cells were lysed in RIPA buffer (50 mM Tris-HCl pH = 7.4, 150 mM NaCl, 0.1% SDS, 0.5% sodium deoxycholate, 1% Triton X-100, 1x cOmplete protease inhibitor cocktail) for 15 min at 4°C on a rocker shaker. After centrifugation at 14,000 *g* for 10 min, 1 mL of cell lysate was mixed with 50 µL of streptavidin-agarose (Pierce) and incubated at 4°C for 4 hr on a rotary shaker. Resin was washed twice with 1 mL of RIPA buffer, once with 1 M KCl, once with 0.1 M Na$_2$CO$_3$, once with 2 M urea in 25 mM Tris-HCl pH = 8.0, and twice with RIPA buffer. Proteins were eluted by boiling beads in 50 µL of 2× SDS Laemmli loading buffer plus 2 mM biotin. Eluted proteins were analyzed by western blot.

## Induction of UPR

For UPR experiments, HEK293T cells were cultured in 6-well plates. Culture medium was replaced with complete medium, and thapsigargin (Tg) (Calbiochem) was added at 1 µM concentration for the indicated periods of time. After that, culture medium was removed, and cells were processed for western blot analysis. For western blot analysis, cells were lysed in cold RIPA lysis buffer and incubated at 4°C for 10 min. Lysates were centrifuged at 14,000 *g* for 10 min, and supernatants were collected and mixed with 6× SDS Laemmli buffer.

## Western blot analysis

All protein samples were boiled at 95°C for 10 min, proteins were separated by SDS-PAGE and analyzed by western blotting with the following antibodies: anti-phospho-IRE1α (NB100-2323, 1:10,000, Novus Biologicals), anti-IRE1α (14C10, 1:1000, Cell Signaling), anti-XBP1s (E9V3E, 1:1000, Cell Signaling), anti-tubulin (E7, 1:500, DSH), anti-APX2 (HRP) (ab192968, 1:1000, Abcam), streptavidin-HRP (7403, 1:20,000, Abcam), anti-S1R (B-5, 1:300, Santa Cruz Biotechnology), anti-His tag (HIS.H8, 1:1000, Millipore), anti-calreticulin (ab2907, 1:5000, Abcam), anti-STIM1 (4916, 1:1000, Cell Signaling), and anti-IP$_3$R1 (1:1000, produced in our lab). The HRP-conjugated anti-rabbit (111-035-144, 1:3000) and anti-mouse (115-035-146, 1:3000) secondary antibodies were from Jackson ImmunoResearch.

## Quantification and statistical analyses

To calculate the Mander's colocalization coefficient between the ER (labeled with mCherry-Sec61β construct) and S1R-GFP proteins, JACOP plugin for ImageJ plugin was used (**Bolte and Cordelières, 2006**). Acquisition parameters (such as laser power and gain) were kept constant for different samples. First, background subtraction was performed, and threshold was adjusted manually for S1R-GFP and mCherry-Sec61β channels. Thresholding was performed such as all the ER was segmented in the mCherry-Sec61β channel and S1R microdomains were clearly resolved in the S1R-GFP channel (typically, at 1.5× mean intensity level). For calculating mCherry-Sec61β:TOM20 and S1R-GFP:TOM20 colocalization in expanded samples, Colocalization Highlighter plugin was used (**Collins, 2007**). Threshold levels were selected as mean signal intensities for mCherry-Sec61β and TOM20 channels, and manually adjusted for S1R-GFP channel to clearly segment microdomains (typically, at 1.5× mean intensity level). The area of colocalizaiong between two channels was normalized to the total area occupied by mCherry-Sec61β or S1R-GFP at the same threshold level. For MAPPER colocalization studies, mCherry-MAPPER channel was used for thresholding and identification of ER-PM junctions and used as mask for calculating integral intensity of S1R-GFP in these areas. Threshold level was adjusted until MAPPER puncta were clearly separated from the background. The mask was saved and used for calculation total S1R-GFP intensity in the masked areas for each cell. These value was normalized to the total fluorescent intensity of the cell to calculate the fraction of S1R residing in MAPPER-positive puncta.

For DSLB quantification, fluorescent intensities of individual LAT-647 and S1R molecules were manually measured after background subtraction in ImageJ. Similarly, the mean intensity of mono-cysteine-labeled LAT-647 molecules was measured in a separate experiment using the same laser power and gain settings. Intensities of small (putative trimers) S1R oligomers were divided by the calculated mean intensity of the pLAT-647 molecule to convert them to a number of molecules per cluster. For quantification of the domain formation in DSLBs, fluid bilayer areas were examined. Bilayer defects or unwashed liposomes were omitted from the analysis. To calculate the fraction of protein residing in clusters, we used an in-house MATLAB script. For each field of view, thresholding was applied to identify spots above the background. The threshold level was typically set at the intensity level of five mono-cysteine-labeled LAT-647 molecules. Then, individual intensities of each spot were calculated, local background signal was subtracted, and them summed together to give the value of total protein clustered in each field of view. Integral intensity of S1R clusters in each field of view was divided by the total protein intensity in the same field of view. Individual intensities of protein clusters were converted to the number of fluorescent molecules by dividing each value by the mean intensity of LAT-647 as described for the smaller oligomers above. For future analysis, we considered clusters as protein assemblies with more than 10 molecules of S1R. Size distribution data was considered lognormal based on Kolmogorov-Smirnov test. For molecular weight distributions, calculated cluster sizes from at least ten fields of view were pooled together and used for calculation of the frequency histogram (bin width = 2) using GraphPad Prism 8. Frequency distribution was interpolated assuming lognormal fit of the data. For cluster size calculations, median cluster size was calculated for each experimental condition.

For western blot analysis, data were densitometrically analyzed using ImageJ software by normalizing the density of each band to IRE1α (for p-IRE1α quantification) or tubulin (for XBP1s quantification) of the same sample after background subtraction. The number of individual experiments, number of total cells or fields of view analyzed, and significance are reported in the figure legends. Statistical significance was calculated by Student's t-test (for two-group comparison), one-way ANOVA with Tukey's multiple comparison test (for comparison between different experimental groups), or Dunnett's multiple comparison test (for comparison of multiple groups with one control group). The multiplicity-adjusted p-value is reported. $p > 0.05$ = n.s., $*p < 0.05$, $**p < 0.01$, $***p < 0.001$, and $****p < 0.0001$.

## Acknowledgements

We are thankful to Ms Anna Starokadomska for help and assistance with these studies, Dr Meewhi Kim for useful discussions, Dr Daniel Ryskamp for generation of S1R KO HEK239 cell line, Dr Arun Radhakrishnan for comments on the manuscript, and Dr Elena Vasileva for support. We are grateful to Drs Michael Hayden and Michal Geva (Prilenia) for encouraging and supporting our work on S1R.

JL is a Sowell Family Scholar in Medial Research. IB holds the Carl J. and Hortense M. Thomsen Chair in Alzheimer's Disease Research. This work was supported by the Russian Science Foundation Grant 19-15-00184 (IB) and by the grants form the National Institutes of Health R01NS056224 (IB), R01AG055577 (IB), and R01GM113079 (JL). MR is supported by the Howard Hughes Medical Institute and a grant from the Welch Foundation (I-1544).

## Additional information

### Funding

| Funder | Grant reference number | Author |
|---|---|---|
| National Institutes of Health | R01NS056224 | Ilya Bezprozvanny |
| National Institutes of Health | R01AG055577 | Ilya Bezprozvanny |
| National Institutes of Health | R01GM113079 | Jen Liou |
| Howard Hughes Medical Institute | | Michael K Rosen |
| Welch Foundation | | Michael K Rosen |
| Russian Science Foundation | 19-15-00184 | Ilya Bezprozvanny |

The funders had no role in study design, data collection and interpretation, or the decision to submit the work for publication.

### Author contributions

Vladimir Zhemkov, Conceptualization, Data curation, Formal analysis, Validation, Investigation, Visualization, Methodology, Writing - original draft, Writing - review and editing; Jonathon A Ditlev, Resources, Investigation, Visualization, Methodology, Writing - review and editing; Wan-Ru Lee, Resources, Investigation, Visualization, Methodology; Mikaela Wilson, Validation, Investigation, Visualization, Methodology; Jen Liou, Resources, Supervision, Funding acquisition, Investigation, Visualization, Writing - review and editing; Michael K Rosen, Resources, Supervision, Funding acquisition, Investigation, Writing - review and editing; Ilya Bezprozvanny, Conceptualization, Resources, Data curation, Formal analysis, Supervision, Funding acquisition, Investigation, Visualization, Writing - original draft, Project administration, Writing - review and editing

### Author ORCIDs

Vladimir Zhemkov (iD) https://orcid.org/0000-0002-6554-1938
Jonathon A Ditlev (iD) https://orcid.org/0000-0001-8287-7700
Jen Liou (iD) http://orcid.org/0000-0003-1546-3115
Michael K Rosen (iD) http://orcid.org/0000-0002-0775-7917
Ilya Bezprozvanny (iD) https://orcid.org/0000-0001-7006-6951

### Decision letter and Author response

Decision letter https://doi.org/10.7554/eLife.65192.sa1
Author response https://doi.org/10.7554/eLife.65192.sa2

## Additional files

### Supplementary files

• Transparent reporting form

### Data availability

All data generated or analysed during this study are included in the manuscript and supporting files. Raw data and tables used for generation of the figures are available at https://doi.org/10.5061/dryad.9zw3r22dn.

The following dataset was generated:

| Author(s) | Year | Dataset title | Dataset URL | Database and Identifier |
|---|---|---|---|---|
| Bezprozvanny I, Zhemkov V, Ditlev JA, Lee W-R, Wilson M, Liou J, Rosen MK | 2021 | The role of sigma-1 receptor in organization of endoplasmic reticulum signaling microdomains | https://doi.org/10.5061/dryad.9zw3r22dn | Dryad Digital Repository, 10.5061/dryad.9zw3r22dn |

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

# Appendix 1

**Appendix 1—key resources table**

| Reagent type (species) or resource | Designation | Source or reference | Identifiers | Additional information |
|---|---|---|---|---|
| Gene (*Homo sapiens*) | *SIGMAR1* | GenBank | NM_005866 | |
| Gene (*synthetic gene*) | TM17 | GenScript | | AAGAAATGG TGGGCTGCG GCGCTGTTA GCTGCGGCG TTACTGGCTG CGGCGTGGT GGAAGAAATAA |
| Gene (*synthetic gene*) | TM27 | GenScript | | AAGAAATGG TGGGCTGCG GCGCTGTTAG CTGCGGCGTT ACTGGCTGCG GCGCTGCTTG CTGCGGCGTT ATTAGCTGCG GCGTGGTGGA AGAAATAA |
| Gene, fragment (*Homo sapiens*) | IRE1a-TM | GenScript | | GCGCGTCCGG AGGCGCCGGT GGACAGCATGC TGCAGGATATG GCGACCATCAT TCTGAGCACCT TCCTGCTGATC GGTTGGGTTGC GTTTATCATTAC CTACCCGCTGA GCAAG |
| Strain, strain background (*Escherichia coli*) | BL21(DE3) | Thermo Fisher | C600003 | Chemical competent cells |
| Cell line (*Homo sapiens*) | HEK293T | ATCC | 3216 | |
| Cell line (*Homo sapiens*) | HEK293 S1R KO | This study | | Generated by co-transfection with Cas9 and guide plasmids, following by antibiotic selection with puromycin and blasticidin |
| Biological sample (*Mus musculus*) | Liver | The Jackson Laboratory | 000664 | Freshly isolated from C57/B6 mouse strain |
| Antibody | Anti-phospho-IRE1a (rabbit polyclonal) | Novus Biologicals | NB100-2323 | WB (1:10,000) |
| Antibody | Anti-IRE1a (rabbit monoclonal) | Cell Signaling | 14C10 | WB (1:1000) |
| Antibody | Anti-XBP1s (rabbit monoclonal) | Cell Signaling | E9V3E | WB (1:1000) |
| Antibody | Anti-tubulin (mouse monoclonal) | DSHB | E7 | WB (1:500) |
| Antibody | Anti-APX2-HRP (rabbit polyclonal) | Abcam | ab192968 | WB (1:1000) |

*Continued on next page*

*Appendix 1—key resources table continued*

| Reagent type (species) or resource | Designation | Source or reference | Identifiers | Additional information |
|---|---|---|---|---|
| Antibody | Anti-S1R (mouse monoclonal) | Santa Cruz | B-5 | WB (1:300) |
| Antibody | Anti-His-tag (mouse monoclonal) | Millipore | HIS.H8 | WB (1:1000) |
| Antibody | Anti-STIM1 (rabbit polyclonal) | Cell Signaling | 4916 | WB (1:1000) |
| Antibody | Anti-calreticulin (rabbit polyclonal) | Abcam | Ab2907 | WB (1:5000) |
| Antibody | Anti-IP3R1 (rabbit polyclonal) | Home made | | WB (1:1000) |
| Antibody | Anti-TOM20 (rabbit monoclonal) | Cell Signaling | D8T4N | WB (1:1000) IF (1:250) |
| Antibody | Anti-GFP (chicken polyclonal) | Abcam | ab13970 | IF (1:1000–1:500) |
| Antibody | Anti-mCherry (rat monoclonal) | Invitrogen | 16D7 | IF (1:1000–1:500) |
| Antibody | Anti-TOM20 (rabbit polyclonal) | Santa Cruz | FL-145 | IF (1:500) |
| Antibody | Anti-mouse-HRP (goat polyclonal) | Jackson ImmunoResearch | 115-035-146 | WB (1:3000) |
| Antibody | Anti-rabbit-HRP (goat polyclonal) | Jackson ImmunoResearch | 115-035-144 | WB (1:3000) |
| Antibody | Anti-chicken-488 (goat polyclonal) | Invitrogen | A11039 | IF (1:1000) |
| Antibody | Anti-rat-594 (donkey polyclonal) | Invitrogen | A21209 | IF (1:1000) |
| Antibody | Anti-rabbin-Atto647N (goat polyclonal) | Sigma-Aldrich | 40839 | IF (1:1000) |
| Recombinant DNA reagent | pFast-Bac-HTA | Invitrogen | 10584027 | |
| Recombinant DNA reagent | Lentivector | Addgene | 14883 | |
| Recombinant DNA reagent | pEGFP-N2 | Clontech | PR29968 | |
| Recombinant DNA reagent | LentiGuide-Puro | Addgene | 52963 | |
| Recombinant DNA reagent | Lenti-Cas9-Blast | Addgene | 52962 | |
| Recombinant DNA reagent | S1R-GFP | This study | | Fusion gene of S1R and GFP in pEGFP-N2 vector |
| Recombinant DNA reagent | S1R-R7ER8E-GFP | This study | | Fusion gene of S1R and GFP in pEGFP-N2 vector with introduced R7ER8E mutation |
| Recombinant DNA reagent | S1R-ΔRR-GFP | This study | | Fusion gene of S1R and GFP in pEGFP-N2 vector with introduced ΔRR mutation |

*Continued on next page*

*Appendix 1—key resources table continued*

| Reagent type (species) or resource | Designation | Source or reference | Identifiers | Additional information |
|---|---|---|---|---|
| Recombinant DNA reagent | S1R-4G-GFP | This study | | Fusion gene of S1R and GFP in pEGFP-N2 vector with introduced 4G mutation |
| Recombinant DNA reagent | S1R-4A-GFP | This study | | Fusion gene of S1R and GFP in pEGFP-N2 vector with introduced 4A mutation |
| Recombinant DNA reagent | S1R-W9L/W11L-GFP | This study | | Fusion gene of S1R and GFP in pEGFP-N2 vector with introduced W9L/W11L mutations |
| Recombinant DNA reagent | S1R-Y173S-GFP | This study | | Fusion gene of S1R and GFP in pEGFP-N2 vector with introduced Y173S mutation |
| Recombinant DNA reagent | S1R-Y201S/Y206S-GFP | This study | | Fusion gene of S1R and GFP in pEGFP-N2 vector with introduced Y201S/Y206S mutations |
| Recombinant DNA reagent | S1R-Δ4-GFP | This study | | Fusion gene of S1R and GFP in pEGFP-N2 vector with introduced Δ4 mutation |
| Recombinant DNA reagent | mCherry-Sec61b | Addgene | 49155 | Zurek et al. Traffic. 2011 |
| Recombinant DNA reagent | mCherry-MAPPER | This study | | mCherry-MAPPER was generated by replacing GFP with mCherry in the original construct from Chang et al. Cell Reports, 2013 |
| Recombinant DNA reagent | His-S1R | This study | | His-tagged S1R in pFastBac-HTA vector |
| Recombinant DNA reagent | His-S1R-4G | This study | | His-tagged S1R in pFastBac-HTA vector with introduced 4G mutation |
| Recombinant DNA reagent | His-S1R-W9L/W11L | This study | | His-tagged S1R in pFastBac-HTA vector with introduced W9L/W11L mutations |
| Recombinant DNA reagent | S1R-6His | This study | | His-tagged S1R in lentivector |
| Recombinant DNA reagent | S1R-4G-6His | This study | | His-tagged S1R in lentivector with introduced 4G mutation |
| Recombinant DNA reagent | S1R-W9L/W11L-6His | This study | | His-tagged S1R in lentivector with introduced W9L/W11L mutations |
| Recombinant DNA reagent | MBP-TM17 | This study | | Generated by cloning TM17 into pMAL-c5x vector |

*Continued on next page*

*Appendix 1—key resources table continued*

| Reagent type (species) or resource | Designation | Source or reference | Identifiers | Additional information |
|---|---|---|---|---|
| Recombinant DNA reagent | MBP-TM27 | This study | | Generated by cloning TM27 into pMAL-c5x vector |
| Recombinant DNA reagent | MBP-IRE1a-TM | This study | | Generated by cloning IRE1a-TM into pMAL-c5x vector |
| Recombinant DNA reagent | S1R-APEX2 | This study | | Generated by cloning S1R-APEX2 fusion gene into lentivector |
| Recombinant DNA reagent | Sec61b-APEX2 | This study | | Generated by cloning Sec61b-APEX fusion gene (RRID:Addgene_83411) into lentivector |
| Recombinant DNA reagent | APEX2-KDEL | This study | | Generated by adding KDEL-encoding sequence to the APEX2 gene and cloning into lentivector |
| Sequence-based reagent | Guide sequence targeting exon 1 of human S1R | This study | | GCGCGAAGAG ATAGCGCAGT |
| Sequence-based reagent | Guide sequence targeting LacZ | Platt et al. Cell, 2014 | | GTGCGAATAC GCCCACGCGAT |
| Commercial assay or kit | Q5 site-directed mutagenesis kit | NEB | E0554 | |
| Chemical compound, drug | 18:1 PC (DOPC) | Avanti Polar Lipids | 850375 | |
| Chemical compound, drug | 24:1 PC (DNPC) | Avanti Polar Lipids | 850399 | |
| Chemical compound, drug | NBD-PC | Avanti Polar Lipids | 810130 | |
| Chemical compound, drug | Cholesterol | Avanti Polar Lipids | 700000 | |
| Chemical compound, drug | DSPE-PEG(2000)-biotin | Avanti Polar Lipids | 880129 | |
| Chemical compound, drug | DGS-NTA-Ni | Avanti Polar Lipids | 790404 | |
| Chemical compound, drug | Egg PC | Avanti Polar Lipids | 241601 | |
| Chemical compound, drug | LDAO | Anatrace | D360 | |
| Chemical compound, drug | OG | Anatrace | O311 | |
| Chemical compound, drug | DDM | Anatrace | D310 | |
| Chemical compound, drug | Cholesteryl hemisuccinate | Anatrace | CH210 | |
| Chemical compound, drug | CarboxyLink agarose | Thermo Scientific | 20266 | |
| Chemical compound, drug | Alexa647 NHS ester | Invitrogen | A37573 | |
| Chemical compound, drug | Alexa555 NHS ester | Invitrogen | A37571 | |

*Continued on next page*

*Appendix 1—key resources table continued*

| Reagent type (species) or resource | Designation | Source or reference | Identifiers | Additional information |
|---|---|---|---|---|
| Chemical compound, drug | C2 maleimide Alexa647 | Invitrogen | A20347 | |
| Chemical compound, drug | (+)-SKF-10047 | Sigma-Aldrich | A114 | |
| Chemical compound, drug | NE-100 hydrochloride | Tocris | 3133 | |
| Software, algorithm | MATLAB_R2019b | MATLAB | | |
| Software, algorithm | GraphPad Prism 8 | GraphPad | | |
| Other | Liposomal mini-extruder with 0.1 mkm polycarbonate filters | Avanti Polar Lipids | 610000 | |

