## [Decision Letter]

**Acceptance summary:**

The reviewers appreciated the manuscript as an advance in understanding sigma-1 receptor biology. The connection between sigma-1 receptor function and membrane sterol content has been alluded to by prior literature, but your manuscript provides one of the first thorough mechanistic investigations of this issue. The link between cholesterol levels and sigma-1 receptor clustering is particular interesting, and lays a foundation for future work in this field.

**Decision letter after peer review:**

Thank you for submitting your article "The role of sigma-1 receptor in organization of endoplasmic reticulum signaling microdomains" for consideration by *eLife*. Your article has been reviewed by 3 peer reviewers, one of whom is a member of our Board of Reviewing Editors, and the evaluation has been overseen by David Ron as the Senior Editor. The following individual involved in review of your submission has agreed to reveal their identity: Wayne D Bowen (Reviewer #3).

The reviewers have discussed the reviews with one another and the Reviewing Editor has drafted this decision to help you prepare a revised submission.

Summary:

This manuscript describes a detailed investigation of the sigma-1 receptor, with an emphasis on the effects of membrane cholesterol content. The authors report that sigma-1 receptor clusters in cholesterol-rich microdomains in the endoplasmic reticulum (ER), contributing to its previously-described localization at mitochondria-associated ER membranes. A series of reconstitution experiments show cholesterol-dependent clustering of the sigma-1 receptor, an effect which is modulated by membrane thickness and drug-like ligands of the receptor. These findings are supplemented by an investigation of the effects of sigma-1 receptor on IRE1a signaling, leading to the finding that sigma-1 knockout attenuates IRE1a function.

Essential revisions:

The reviewers agreed that the manuscript was likely to be of broad interest and addresses important biological questions surrounding the poorly understood sigma-1 receptor. However, concerns were raised regarding a number of points that need to be addressed in order for the manuscript to be suitable for publication. Specifically:

Most of the imaging experiments throughout the manuscript are interpreted only qualitatively, and many of these show relatively minor differences. Objective quantitative analysis should be provided wherever possible. Any subjective assessments should be conducted using blinding to avoid introduction of bias.

The connection between the biological effects on IRE1a activation and cholesterol-dependent clustering is relatively indirect. The reviewers agree that additional experimental data should be provided to further assess the validity of the authors' proposed model. For example, inclusion of rescue experiments in sigma-1 knockout cells using the cholesterol-binding mutants would help to strengthen the connection between IRE1a function and membrane cholesterol content. Similarly, disruption of cholesterol-rich domains by addition of beta-cyclodextrin could provide additional evidence to support the model. In addition, testing the effects of ligands in the cellular imaging experiments would strengthen the link between in vitro biophysical experiments and cellular physiology.

A related issue is that cholesterol binding is not tested explicitly for certain sigma-1 receptor mutants, potentially confounding interpretation of experimental data. These include experiments where alterations were made to the S1R sequence, with results interpreted in light of S1R no longer being able to bind cholesterol. Two specific places where this issue arises are:

1. Studies described on pages 6-7 and shown in Figure 3B where wild-type sigma-1 receptor is compared to S1R-Y201S/Y206S, S1R-Y173S, S1R-4G, and S1R-W9L/W11L. These mutations had differential effects on receptor distribution that were attributed to alterations in cholesterol binding without confirming the changes in cholesterol binding. This is particularly relevant for the explanation given for why S1R-W9L/W11L fails to cluster in both cells and the cholesterol supplemented GUV system, while the S1R-4G mutant exhibited cholesterol-induced clustering in the GUV system but not in cells (page 7, lines 27-31).

2. Another example is the membrane thickness experiment described at the top of page 8 and shown in Figure 4A. Shortening the S1R by deletion of 4 aa in the TM region produced a sigma-1 receptor that exhibited a more diffuse distribution when expressed in HEK293 cells. The authors appear to be attributing this only to the decreased length of the sigma-1 receptor transmembrane domain. However, it seems feasible (based on their other data) that if this construct fails to bind cholesterol, the same result would be observed. Confirming that the truncated sigma-1 receptor does in fact bind cholesterol would strengthen the argument being made here.

---

## [Author Response]

Essential revisions:The reviewers agreed that the manuscript was likely to be of broad interest and addresses important biological questions surrounding the poorly understood sigma-1 receptor. However, concerns were raised regarding a number of points that need to be addressed in order for the manuscript to be suitable for publication. Specifically:Most of the imaging experiments throughout the manuscript are interpreted only qualitatively, and many of these show relatively minor differences. Objective quantitative analysis should be provided wherever possible. Any subjective assessments should be conducted using blinding to avoid introduction of bias.

We quantified the results and repeated evaluation of the data in the blinded fashion.

The connection between the biological effects on IRE1a activation and cholesterol-dependent clustering is relatively indirect. The reviewers agree that additional experimental data should be provided to further assess the validity of the authors' proposed model. For example, inclusion of rescue experiments in sigma-1 knockout cells using the cholesterol-binding mutants would help to strengthen the connection between IRE1a function and membrane cholesterol content. Similarly, disruption of cholesterol-rich domains by addition of beta-cyclodextrin could provide additional evidence to support the model. In addition, testing the effects of ligands in the cellular imaging experiments would strengthen the link between in vitro biophysical experiments and cellular physiology.We agree and attempted S1R rescue experiments in HEK293 S1R KO cells using lentiviruses encoding wild type, 4G and W9L/W11L mutants of S1R. However, S1R expression levels in these experiments were much lower when compared to endogenous levels (evaluated by Western blot, data not shown). Increase in viral titer or addition of polybrene did not considerably improve expression levels of S1R. As a control, we have confirmed efficient transduction with these viruses in primary hippocampal neuronal cultures, indicating that problems with expression are related to the HEK293 cell lines. Because of inability to achieve high levels of S1R expression, we have not been able to perform IRE1a rescue experiments. Impaired IRE1a signaling in S1R KD cells has already been shown previously (Mori et al., PLoS One, 2013). In the absence of S1R rescue experiments, we decided to move the original Figure 6 to supplementary (now Figure 5S2) and focus on biophysical results with IRE1a transmembrane domain which are now included in combined Figure 5 as panels 5G and Figure 5H.A related issue is that cholesterol binding is not tested explicitly for certain sigma-1 receptor mutants, potentially confounding interpretation of experimental data. These include experiments where alterations were made to the S1R sequence, with results interpreted in light of S1R no longer being able to bind cholesterol. Two specific places where this issue arises are:1. Studies described on pages 6-7 and shown in Figure 3B where wild-type sigma-1 receptor is compared to S1R-Y201S/Y206S, S1R-Y173S, S1R-4G, and S1R-W9L/W11L. These mutations had differential effects on receptor distribution that were attributed to alterations in cholesterol binding without confirming the changes in cholesterol binding. This is particularly relevant for the explanation given for why S1R-W9L/W11L fails to cluster in both cells and the cholesterol supplemented GUV system, while the S1R-4G mutant exhibited cholesterol-induced clustering in the GUV system but not in cells (page 7, lines 27-31).To address this question, we performed series of pulldown experiments with cholesteryl-agarose. Recombinant S1R-6*His (WT or mutants) was mixed with cholesterol-coupled resin, washed and analyzed using Western blot analysis. First, we confirmed that recombinant S1R-6*His binds to cholesteryl-agarose, but not to control resin (Figure 4S1). Then we tested binding of S1R-4G-6*His and S1R-W9L/W11L-6*His and concluded that both have reduced affinity for cholesterol (Figure 4C and Figure 4D). To confirm these findings, we expressed exogenous S1R-6*His in HEK293T cells and used cell lysates in similar pulldown experiments. Cells were lysed and lysates were incubated with cholesteryl-agarose. We found that in this assay S1R-W9L/W11L-6*His binds weaker to cholesteryl-agarose compared to wild type protein (Figure 4E and Figure 4F). Binding of S1R-4G-6*His was compatible to the wild type (Figure 4F and Fig4E). Taken together, we concluded that affinity for cholesterol was reduced for both studied mutants, but these effects were more pronounced for S1R-W9L/W11L than for 4G mutant.2. Another example is the membrane thickness experiment described at the top of page 8 and shown in Figure 4A. Shortening the S1R by deletion of 4 aa in the TM region produced a sigma-1 receptor that exhibited a more diffuse distribution when expressed in HEK293 cells. The authors appear to be attributing this only to the decreased length of the sigma-1 receptor transmembrane domain. However, it seems feasible (based on their other data) that if this construct fails to bind cholesterol, the same result would be observed. Confirming that the truncated sigma-1 receptor does in fact bind cholesterol would strengthen the argument being made here.It is possible that delta 4 affects cholesterol binding, but the main rational for importance of membrane thickness comes from S1R/MBP-TM27 co-reconstitution experiments, not from S1R-∆4 mutant. The text related to Figure 4 was edited accordingly.